# SUPER ROBOT VIEW TRANSFORMER

## ABSTRACT

Learning a single model for multiple robotic manipulation tasks, particularly high-precision tasks, has been a long-standing challenge in robotics research due to uncertainties inherent in both the model and the data. These uncertainties, namely *epistemic uncertainty* arising from model limitations and *aleatoric uncertainty* stemming from data variability, hinder precise control. While the Robot View Transformer (RVT) improves performance by re-rendering point clouds from fixed viewpoints and processing structured 2D virtual images, it still suffers from occlusion artifacts in rendering and limited action precision due to resolution constraints. To address these limitations, we propose the Super Robot View Transformer (S-RVT) framework, which integrates three novel components: the Super Point Renderer (S-PR), the Super-resolution Multi-View Transformer (S-MVT), and the Hierarchical Sampling Policy (HSP). The S-PR enhances the rendering process to mitigate occlusion artifacts, while the S-MVT integrates super-resolution to the output heatmaps, enabling finer-grained manipulation. The HSP efficiently samples multi-view heatmaps in 3D space to obtain accurate 3D poses. These innovations collaboratively mitigate the challenges of occlusion and precision in manipulation tasks. Our experimental results demonstrate that S-RVT achieves a success rate of 87.8 % across 18 manipulation tasks, surpassing the state-of-the-art of 81.4 %. Notably, for high-precision manipulation tasks, S-RVT exhibits nearly a two-fold improvement over existing methods, underscoring its effectiveness in precise control scenarios. Our code and trained models will be released to support further research.

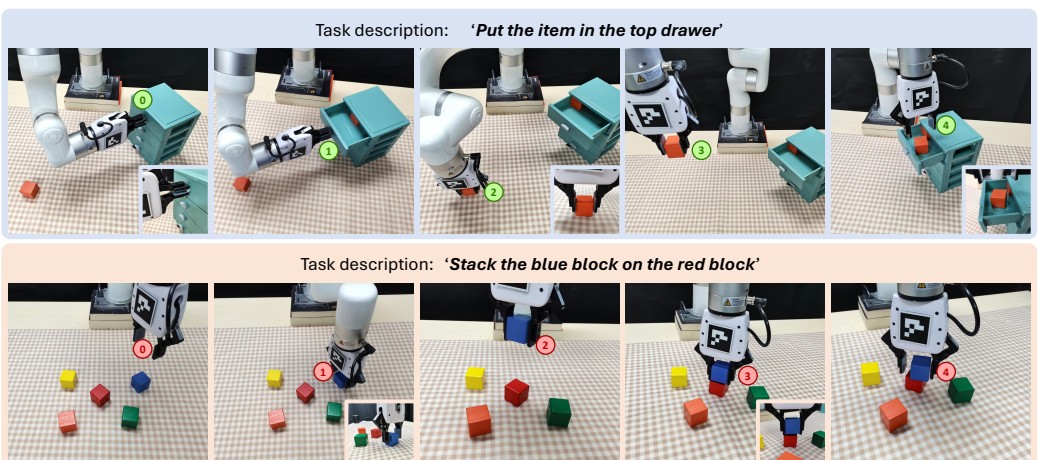

Figure 1: Illustration of S-RVT executing high-precision tasks. Given a natural language task description, a single S-RVT model can perform multiple 3D manipulation tasks with remarkable accuracy. For instance, in the task of *put the item in the top drawer*, the gripper's size is larger compared to the drawer handle, requiring millimeter-level precision for successful grasping.

## 1 INTRODUCTION

Learning diverse manipulation tasks is fundamental and necessary for building intelligent embodied agents. Multi-task learning in robotic manipulation has drawn significant interest in the community

and made considerable progress, thanks to large-scale Imitation Learning (IL) demonstrations and Reinforcement Learning (RL) simulators James et al. (2020); Mees et al. (2022); Yu et al. (2020); Gu et al. (2023); Li et al. (2023); Gong et al. (2023); Padalkar et al. (2023); Wang et al. (2024). These simulators and datasets fuel the development of Embodied Artificial Intelligence (EAI) by enabling researchers to build and test algorithms in controlled environments that closely mimic real-world conditions. In this work, we investigate the problem of multi-task imitation learning for robot manipulation.

Multi-task Imitation Learning challenges robots to acquire diverse skills described by natural language through human demonstrations. This approach encompasses tasks requiring semantic scene understanding (*e.g., take the steak off the grill*), high-precision (*e.g., put the ring on the azure spoke*) or long-horizon planning (*e.g., put the item in the top drawer*). Different from previous methods like Diffusion Policy (DP) Chi et al. (2023), Action Chunking with Transformers (ACT) Zhao et al. (2023) and their variations Fu et al. (2024); Ze et al. (2024); Ke et al. (2024); Ha et al. (2023); Chen et al. (2023a); Xian et al. (2023), which learn task-specific policies, multi-task learning aims to develop a single policy for various tasks. Multi-task Imitation Learning generally falls into two categories: key-state based methods Shi et al. (2023); Xian et al. (2023); Gervet et al. (2023); Goyal et al. (2023; 2024); Shridhar et al. (2023); Chen et al. (2023b) and key-trajectory based methods Ke et al. (2024); Brohan et al. (2022; 2023); Huang et al. (2023). Key-state based methods decompose demonstrations into critical discrete key poses, teaching the robot to predict the subsequent key pose based on current observations. Key pose typically represents the gripper's translation, rotation, and state. Key-trajectory methods directly imitate the demonstrated trajectories. This study focuses on key-state based methods for robotic manipulation.

Previous methods learn the key pose distribution from colored point cloud observation. PerAct Shridhar et al. (2023) employs a voxel-based representation to predict the next key pose, while Act3D Gervet et al. (2023) introduces multi-resolution 3D feature field. Other methods, such as Robot View Transformer (RVT) Goyal et al. (2023), tackle this problem by predicting multi-view heatmaps from rendered RGB-D observations. These approaches have demonstrated impressive results on training and inference speed, semantic understanding and long-horizon planning. However, they fall short in high-precision tasks like *peg insert* or *shape place*. RVT-2 Goyal et al. (2024) addresses challenging precise manipulation tasks by a *zoom-in* process, and demonstrates promising results in extensive experiments.

We advance beyond previous methods by addressing uncertainties inherent in both the model and the data Kendall & Gal (2017); Kendall et al. (2018). These uncertainties are broadly categorized into *epistemic uncertainty*, arising from limitations in the model, and *aleatoric uncertainty*, stemming from inherent variability in the data. For instance, in the task of *put the ring on the azure spoke*, the former uncertainty refers to the fact that the robot may misjudge the exact position of the spoke due to insufficient or biased training data. The latter uncertainty indicates the robot fails to predict the next key pose due to occlusions. The original RVT framework discretizes the action space; however, such a coarse-grained action space is insufficient for accomplishing high-precision manipulation tasks, which contributes to *epistemic* uncertainty. Additionally, the points of interest on the manipulated objects are often occluded by the robot arm, making it difficult for the model to infer the next key pose based on the current observation, thus increasing *aleatoric* uncertainty. To address these uncertainties, we introduce the *Super Robot View Transformer* (S-RVT), a multi-task framework designed for high-precision manipulation tasks.

Our S-RVT framework comprises three key modules: the Super Point Renderer (S-PR), the Super-resolution Multi-View Transformer (S-MVT), and the Hierarchical Sampling Policy (HSP). The S-PR mitigates *aleatoric uncertainty* by addressing observational uncertainties, particularly catastrophic occlusion, where critical visual obstructions impede task completion. The S-MVT and HSP work together to reduce *epistemic uncertainty*: S-MVT enhances model expressivity by generating super-resolution heatmaps with strong supervision, while HSP samples multi-view heatmaps in 3D space to obtain accurate 3D poses. By integrating these modules, S-RVT effectively reduces both epistemic and aleatoric uncertainties, advancing the state-of-the-art in high-precision robotic manipulation tasks. Notably, our method is a general boosting framework for virtual view-based approaches. Thus, we integrate it with both the RVT Goyal et al. (2023) and RVT-2 Goyal et al. (2024), yielding promising results across 18 challenging tasks in the RLBench benchmark James et al. (2020). For RVT, our S-RVT improves the average success from 0.629 to 0.734. Furthermore, our S-RVT2 achieves a success rate of 0.878, surpassing the state-of-the-art 0.814. In tasks requiring

high-precision, such as *peg insertion*, we establish a remarkable success rate of 0.86, achieving a relative 115 % improvement over the state-of-the-art performance of 0.40. We also demonstrate our method's effectiveness in real world, as illustrated in Figure 1.

## 2 RELATED WORK

In this section, we introduce the related work from three different perspectives: Transformers for Manipulation, Multi-task Learning in Robotics and High-Precision Manipulation.

### 2.1 TRANSFORMERS FOR MANIPULATION

In the domain of robotic manipulation, transformer architectures Vaswani et al. (2017) have gained significant traction, demonstrating their efficacy in enhancing control capabilities Johnson et al. (2021); Chaplot et al. (2021); Clever et al. (2021); Yang et al. (2021). These models excel in processing diverse sensory inputs Dasari & Gupta (2021); Jangir et al. (2022); Kim et al. (2021); Liu et al. (2022); Zhao et al. (2023); Goyal et al. (2023); Singh et al. (2023); Simeonov et al. (2023), leveraging their inherent ability to handle heterogeneous data streams. This multi-modal adaptability has been further expanded through the integration of transformer backbones with diffusion models Team et al. (2024); Chi et al. (2023); Ze et al. (2024); Xian et al. (2023), facilitating complex, long-horizon motion planning. Notably, transformer-based approaches have shown remarkable prowess in feature extraction from various input modalities. However, a recurring theme in existing literature is the necessity for extensive training datasets, typically encompassing hundreds of task-specific demonstrations, to achieve robust performance. This data-intensive requirement underscores the challenges associated with transformer deployment in robotic systems.

### 2.2 MULTI-TASK LEARNING IN ROBOTICS

Recent years have witnessed growing interest in the robotics community towards developing models capable of performing multiple tasks using a single model. Some researchers have focused on achieving multi-task generalization by learning generalizable task or action representations Brohan et al. (2022); Gubbi et al. (2020); Guhur et al. (2023); Kim et al. (2021); Lee et al. (2019). However, the limitations of these representations often restrict generalization to specific task categories. More recent studies have expanded the scope by using language to specify a broader range of tasks, while training policies on large-scale, pre-collected datasets Jang et al. (2022); Rohmer et al. (2013); Schoettler et al. (2020); Shi et al. (2023); Simeonov et al. (2023); Radosavovic et al. (2023b); Nair et al. (2022); Xiao et al. (2022); Radosavovic et al. (2023a); Wu et al. (2024); Black et al. (2024); Gu et al. (2024). Concurrent research efforts have explored various approaches to enhance model capabilities, including learning more effective visual representations through masked image modeling Xiao et al. (2022); Radosavovic et al. (2023b) or contrastive learning Nair et al. (2022); Sermanet et al. (2018), as well as developing world models Seo et al. (2023); Hafner et al. (2020).

### 2.3 HIGH-PRECISION MANIPULATION

High-precision manipulation is crucial in tasks with stringent requirements for action accuracy. To develop high-precision strategies, previous studies have employed various sensory modalities and data-intensive learning algorithms. Initial investigations utilize proprioceptive data for learning peg-in-hole task through imitation learning Gubbi et al. (2020). Subsequently, (Schoettler et al., 2020) and (Tang et al., 2023) apply reinforcement learning algorithms to accomplish insertion tasks using visual sensory input or proprioceptive data. To enhance execution accuracy further, researchers incorporate tactile feedback, including torque sensors Lee et al. (2019); Liu et al. (2020) and vision-based tactile sensors Dong et al. (2021); Xu et al. (2023). However, these approaches rely on algorithms requiring substantial training data (*e.g.*, reinforcement learning or imitation learning from hundreds of demonstrations) and are limited to learning a single model per task.

In contrast to prior methods, we adopt the multi-task learning paradigm of Robot View Transformer (RVT) Goyal et al. (2023), which employs structured representations such as point cloud renderings of virtual views. While RVT enables robots to acquire more robust skills with fewer demonstrations, it falls short in tasks requiring high precision due to uncertainties in both observation and

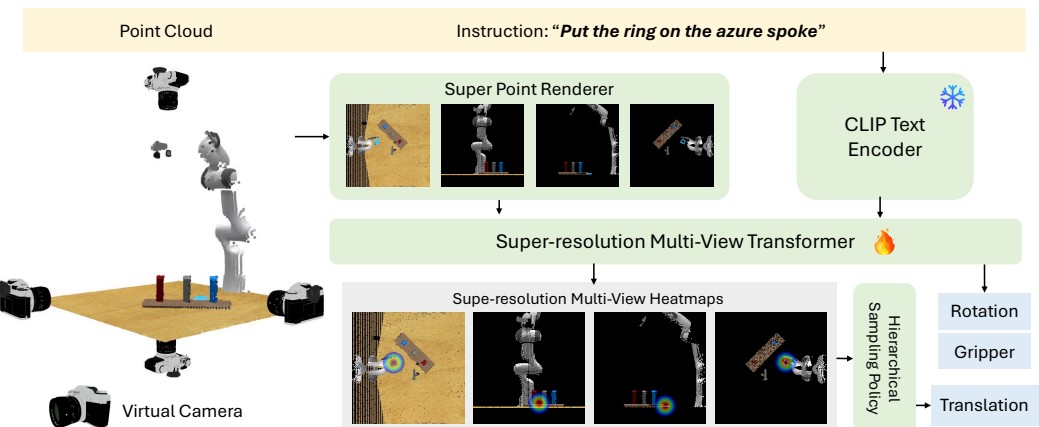

Figure 2: S-RVT framework. S-RVT takes RGB point cloud and natural language task description as input. The process begins with the Super Point Renderer, which projects the 3D point cloud into multi-view images by four virtual cameras of fixed positions. These images, along with language features, are then fed into the Super-resolution Multi-View Transformer. This module outputs high-resolution heatmaps representing the key pose translation in 2D projections, as well as the gripper's rotation and the state for the next key pose. Finally, the Hierarchical Sampling Policy is employed to sample and convert the multi-view 2D heatmaps back into 3D translation coordinates.

action spaces. To address these challenges, we build upon RVT and propose three key innovations aimed at enhancing precision in robotic manipulation tasks: Super Point Renderer, which mitigates occlusions in point cloud renderings; Super-resolution Robot View Transformer, which generates higher-resolution heatmaps for finer-grained operations; and Hierarchical Sampling Policy, which samples multi-view heatmaps in 3D space to obtain precise 3D poses. These enhancements significantly improve our model's performance in high-precision manipulation tasks.

## 3 METHOD

Our approach presents a general boosting strategy for virtual view-based methods. By addressing both epistemic and aleatoric uncertainties, we advance previous virtual view-based methods, including RVT and RVT2, specifically for high-precision manipulation tasks. In this section, we elucidate the overarching concept of our method by focusing on the S-RVT model rather than S-RVT2, because S-RVT2 only adds a *zoom-in* process compared to S-RVT. We first present the problem formulation, followed by a detailed description of our three key strategies: Super Point Renderer, Super-resolution Robot View Transformer, and Hierarchical Sampling Policy. We discuss the loss function at the end of this section.

### 3.1 PROBLEM FORMULATION

Our method falls under the category of key-state based manipulation. In this approach, the trajectory of the robot's end-effector is represented by a sequence of key poses. For instance, given the task description *open the top drawer*, the sequence of key poses can be decomposed into *pre-grasp for the top drawer handle*, *grasp pose*, and *pull pose for the drawer*.

The training set $\mathcal{D}_{\text{train}}$ consists of pairs of task descriptions and corresponding sequences of observations, key poses and gripper states, *i.e.*, $\mathcal{D}_{\text{train}} = \{[l_k, (o_i^{(k)}, \mathbf{T}_i^{(k)}, g_i^{(k)})]_{i=1}^{N_k}\}_{k=1}^{M}$, where $l_k$ is the task description for the $k$-th demonstration, $o_i^{(k)}$ is the RGB point cloud observation at step $i$, $\mathbf{T}_i^{(k)} \in SE(3)$ is the end-effector pose, $g_i^{(k)}$ is the gripper state (open or closed) and $N_k$ is the episode length for the $k$-th demonstration. Our model is trained on these trajectory data to learn a mapping from the current observation and task description to the next key pose and gripper state. Specifically, during training, the model receives the current task description $l_k$ and the current RGB point cloud observation $o_i^{(k)}$, and learns to predict the next key pose $\mathbf{T}_{i+1}^{(k)}$ and gripper state $g_{i+1}^{(k)}$. During testing, the robot uses the trained model to predict the next key pose $\hat{\mathbf{T}}_{i+1} \in SE(3)$ and the

gripper state $\hat{g}_{i+1}$ based on its current task description $l$ and the RGB point cloud observation $o_i$, where $i$ denotes the index of the current step. Once predicted, this pose is input to a motion planner, which generates a trajectory towards it. Upon reaching the predicted pose $\hat{T}_{i+1}$, the system acquires a new point cloud observation and proceeds to predict the subsequent key pose. This iterative process continues until either the success condition is met, or the execution steps exceed a predefined limit, or a collision occurs.

## 3.2 SUPER POINT RENDERER

To enable precise robotic manipulation and address the occlusion and perspective distortion challenges, we propose the Super Point Renderer (S-PR) module. The S-PR transforms RGB-D point cloud observations into 2D virtual images. We implement multi-view rendering from different viewpoints {*top*, *front*, *right*, *down*} to ensure comprehensive scene coverage.

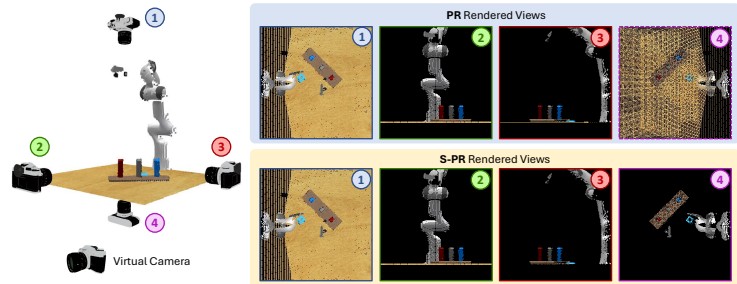

Figure 3: Visualization of the results from Super Point Renderer and Point Renderer Goyal et al. (2023; 2024). The fourth rendered result with dotted box is non-exsitent in original RVT-2 Goyal et al. (2024). We visualize it here for better comparison.

Specifically, we first introduce an occlusion handling policy for the *right* and *down* views. We preprocess the point cloud using CUDA-accelerated DBSCAN clustering in the color space to filter out occluding elements like the tabletop while retaining task-relevant features. The robotic arm is retained as its pose provides valuable information about task progression. Second, we use an orthographic camera model to project the point cloud onto the image plane, preserving geometric relationships without perspective distortion. This rendering process comprises three key steps and is implemented in CUDA for acceleration which is introduced by Goyal et al. (2023). 1) The 3D points are projected onto the 2D image plane by converting them into image coordinates using GPU-accelerated matrix operations. 2) Z-ordering is applied to identify the point with the smallest depth for each pixel. 3) Screen-space splatting is used to model the points as finite-radius discs rather than singular pixels. As illustrated in Figure 3, in tasks such as *put the ring on the azure spoke*, standard point rendering fails to provide clear visual cues due to occlusions, making it difficult for the model to learn necessary rotations. Our S-PR mitigates this problem by multi-view rendering from different viewpoints, enhancing the model's understanding in complex manipulations. In Figure 3, the filtered down view of our S-PR avoids the occlusions and explicitly shows the *azure spoke* and *ring* positions.

## 3.3 SUPER-RESOLUTION MULTI VIEW TRANSFORMER

The task descriptions are processed through the CLIP (ResNet-50) text encoder to extract features, which, together with the rendered multi-view images, are then fed into the Super-resolution Multi-View Transformer (S-MVT). As shown in Figure 2, S-MVT generates super-resolution heatmaps; we denote the super-resolution factor as sr. Additionally, S-MVT outputs the rotation and gripper opening predictions for the next key pose. Specifically, the virtual images and language features are processed through an MVT structure similar to that in RVT, producing feature maps. These feature maps undergo upsampling to produce an sr-fold super-resolution heatmap, representing the probability distribution of possible 3D poses projected onto the 2D plane. Our upsampling employs an Efficient Up-convolution Block (EUCB), which uses Depthwise Separable Convolution (DWC) to reduce computational cost and parameter count while improving output resolution and preserving feature details. To predict the rotation and gripper state, we sample features from the image patch corresponding to the projected 3D position of the predicted next key pose on the virtual view. These sampled features are then processed through an MLP to estimate the rotation and gripper opening. This conditional sampling approach is employed because the gripper's rotation and opening are intrinsically linked to its translation, thereby yielding more plausible predictions. The details of our model architecture are discussed in Appendix A.1.

## 3.4 HIERARCHICAL SAMPLING POLICY

We obtain multi-view heatmaps through S-MVT, where each heatmap represents the probability distribution of possible 3D poses projected onto the 2D plane. Our goal is to predict a precise 3D pose from these multi-view 2D probability distributions. A straightforward approach would be to uniformly distribute particles in 3D space at the resolution of the 2D views, project these particles onto different 2D planes, obtain the corresponding grid probability values, sum the probabilities from different views, and select the particle with the highest cumulative probability as the prediction result. However, for high-resolution heatmaps, uniformly distributing particles in 3D space at increasing resolutions leads to higher particle density, causing GPU memory overflow. To address this

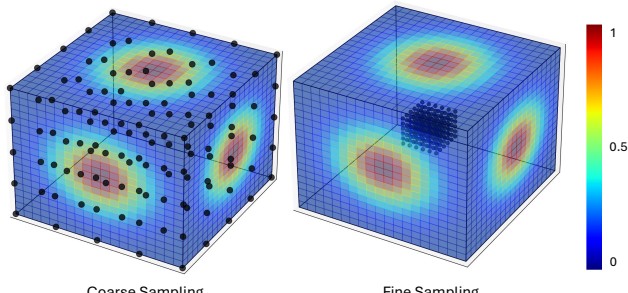

Coarse Sampling          Fine Sampling

Figure 4: Illustration of Hierarchical Sampling Policy. To generate 3D key pose from 2D multi-view heatmaps, we design a coarse-to-fine sampling policy. Initially, sampling points are uniformly distributed in the 3D space. These points are then projected onto each heatmap, and the location with the highest sum of probability values across all views is selected as the coarse prediction. Subsequently, in the fine sampling stage, a more detailed sampling is conducted in the vicinity of this coarse prediction, yielding the final estimation of the key pose translation.

issue, we develop the Hierarchical Sampling Policy. First, we sample at a lower resolution to obtain a coarse predicted pose. Subsequently, we perform a higher-density sampling in the vicinity of this initial prediction to refine the pose estimate, as depicted in Figure 4. This approach enables high-precision operations in robotic manipulation tasks, such as accurately inserting a charging plug into a socket.

## 3.5 LOSS FUNCTION

As illustrated in Figure 2, our model generates three outputs: multi-view heatmaps, rotation of the next key pose, and gripper state. For heatmap supervision, we project the ground-truth (GT) 3D key pose onto various viewpoints to obtain GT heatmaps. While cross-entropy loss is effective for standard heatmaps, it is inadequate for super-resolution heatmaps. We attribute this to the increased resolution (*e.g.*, 1 projected pixel out of $1000^2$ total pixels) making it challenging for the network to focus on sparse, difficult samples. To address this, we implement focal loss Lin (2017), denoted as $l_t$, which emphasizes these challenging samples. The focal loss is defined as:

$$l_t = -\sum[\alpha(1-\hat{h})^\gamma h \log(\hat{h}) + (1-\alpha)\hat{h}^\gamma(1-h)\log(1-\hat{h})], \tag{1}$$

where $\hat{h}$ is the predicted probability of the flattened multi-view heatmaps, $h$ represents the GT 3D translation projected onto multi-view one-hot heatmaps, $\alpha$ is the balancing factor, and $\gamma$ is the focusing parameter. For rotation and gripper state prediction, we formulate these tasks as classification problems and employ cross-entropy loss, denoted as $l_r$ and $l_s$ respectively. Specifically, we supervise the rotation by quantizing Euler angles into discrete bins. The cross-entropy loss for rotation and gripper state is defined as:

$$l_r = -\sum[r\log(\hat{r}) + (1-r)\log(1-\hat{r})], \quad l_s = -\sum[s\log(\hat{s}) + (1-s)\log(1-\hat{s})], \tag{2}$$

where $\hat{r}$ and $\hat{s}$ represent discrete Euler angles and gripper state, $r$ indicates the GT one-hot Euler angles and $s$ denotes the GT gripper state.

Given the disparate pattern of these three prediction tasks, simply summing their losses lead to imbalanced learning. To mitigate this, we employ Uncertainty Weighting Kendall et al. (2018) to balance the learning across different losses. This approach allows the model to learn the optimal weighting for each task by considering the homoscedastic uncertainty associated with them. Specifically, we define the total loss function as:

$$L = \frac{1}{2\sigma_t^2}l_t + \frac{1}{2\sigma_r^2}l_r + \frac{1}{2\sigma_s^2}l_s + \ln\sigma_t + \ln\sigma_r + \ln\sigma_s. \tag{3}$$

| Models | Avg. Success ↑ | Avg. Rank ↓ | Close Jar | Drag Stick | Insert Peg | Meat off Grill | Open Drawer | Place Cups | Place Wine | Push Buttons |
|---|---|---|---|---|---|---|---|---|---|---|
| Image-BC (CNN) Jang et al. (2022) | 1.3 | 9.5 | 0 | 0 | 0 | 0 | 4 | 0 | 0 | 0 |
| Image-BC (ViT) Jang et al. (2022) | 1.3 | 9.7 | 0 | 0 | 0 | 0 | 0 | 0 | 0 | 0 |
| C2F-ARM-BC James et al. (2022) | 20.1 | 8.6 | 24 | 24 | 4 | 20 | 20 | 0 | 8 | 72 |
| HiveFormer Guhur et al. (2023) | 45.3 | 6.9 | 52.0 | 76.0 | 0.0 | **100.0** | 52.0 | 0.0 | 80 | 84 |
| PolarNet Chen et al. (2023b) | 46.4 | 6.3 | 36.0 | 92.0 | 4.0 | **100.0** | 84.0 | 0.0 | 40 | 96 |
| PerAct Shridhar et al. (2023) | 49.4 | 6.1 | 55.2 ± 4.7 | 89.6 ± 4.1 | 5.6 ± 4.1 | 70.4 ± 2.0 | 88.0 ± 5.7 | 2.4 ± 3.2 | 44.8 ± 7.8 | 92.8 ± 3.0 |
| Act3D Gervet et al. (2023) | 65.0 | 4.2 | 92.0 | 92.0 | 27.0 | 94.0 | **93.0** | 3.0 | 80 | 99 |
| RVT Goyal et al. (2023) | 62.9 | 4.4 | 52.0 ± 2.5 | 99.2 ± 1.6 | 11.2 ± 3.0 | 88.0 ± 2.5 | 71.2 ± 6.9 | 4.0 ± 2.5 | 91.0 ± 5.2 | **100.0 ± 0.0** |
| RVT-2 Goyal et al. (2024) | 81.4 | 2.6 | **100.0 ± 0.0** | 99.0 ± 1.7 | 40.0 ± 0.0 | 99.0 ± 1.7 | 74.0 ± 11.8 | 38.0 ± 4.5 | 95.0 ± 3.3 | **100.0 ± 0.0** |
| **S-RVT (ours)** | 73.4 | 2.7 | 77.3 ± 2.1 | **100.0 ± 0.0** | 15.3 ± 4.7 | **100.0 ± 0.0** | 78.0 ± 4.2 | 10.7 ± 3.3 | **97.3 ± 2.1** | **100.0 ± 0.0** |
| **S-RVT2 (ours)** | **87.8** | **1.5** | **100.0 ± 0.0** | **100.0 ± 0.0** | **86.0 ± 2.2** | **100.0 ± 0.0** | 81.3 ± 6.0 | **39.3 ± 7.3** | 95.3 ± 3.0 | **100.0 ± 0.0** |

| Models | Put in Cupboard | Put in Drawer | Put in Safe | Screw Bulb | Slide Block | Sort Shape | Stack Blocks | Stack Cups | Sweep to Dustpan | Turn Tap |
|---|---|---|---|---|---|---|---|---|---|---|
| Image-BC (CNN) Jang et al. (2022) | 0 | 8 | 4 | 0 | 0 | 0 | 0 | 0 | 0 | 8 |
| Image-BC (ViT) Jang et al. (2022) | 0 | 0 | 0 | 0 | 0 | 0 | 0 | 0 | 0 | 16 |
| C2F-ARM-BC James et al. (2022) | 0 | 4 | 12 | 8 | 16 | 8 | 0 | 0 | 0 | 68 |
| HiveFormer Guhur et al. (2023) | 32.0 | 68.0 | 76.0 | 8.0 | 64.0 | 8.0 | 8.0 | 0.0 | 28.0 | 80 |
| PolarNet Chen et al. (2023b) | 12.0 | 32.0 | 84.0 | 44.0 | 56.0 | 12.0 | 4.0 | 8.0 | 52.0 | 80 |
| PerAct Shridhar et al. (2023) | 28.0 ± 4.4 | 51.2 ± 4.7 | 84.0 ± 3.6 | 17.6 ± 2.0 | 74.0 ± 13.0 | 16.8 ± 4.7 | 26.4 ± 3.2 | 2.4 ± 2.0 | 52.0 ± 0.0 | 88.0 ± 4.4 |
| Act3D Gervet et al. (2023) | 51.0 | 90.0 | 95.0 | 47.0 | 93.0 | 8.0 | 12.0 | 9.0 | 92.0 | 94 |
| RVT Goyal et al. (2023) | 49.6 ± 3.2 | 88.0 ± 5.7 | 91.2 ± 3.0 | 48.0 ± 5.7 | 81.6 ± 5.4 | 36.0 ± 2.5 | 28.8 ± 3.9 | 26.4 ± 8.2 | 72.0 ± 0.0 | 93.6 ± 4.1 |
| RVT-2 Goyal et al. (2024) | 66.0 ± 4.5 | 96.0 ± 0.0 | 96.0 ± 2.8 | **88.0 ± 4.9** | 92.0 ± 2.8 | 35.0 ± 7.1 | 80.0 ± 2.8 | 69.0 ± 5.9 | **100.0 ± 0.0** | 99.0 ± 1.7 |
| **S-RVT (ours)** | 60.7 ± 4.7 | **100.0 ± 0.0** | 89.3 ± 2.1 | 54.0 ± 3.3 | **100.0 ± 0.0** | 54.7 ± 5.5 | 72.7 ± 3.3 | 54.7 ± 3.3 | 60.7 ± 1.6 | 96.0 ± 2.5 |
| **S-RVT2 (ours)** | **70.7 ± 4.1** | 98.7 ± 2.0 | **98.0 ± 2.2** | **88.0 ± 2.5** | 84.0 ± 5.1 | **71.3 ± 8.1** | **80.7 ± 5.8** | **90.0 ± 3.3** | 97.3 ± 2.0 | **99.3 ± 1.6** |

Table 1: Comparisons of our S-RVT and S-RVT2 with state-of-the-art methods. S-RVT2 outperforms RVT-2 Goyal et al. (2024) by achieving a 6.4 % higher average success rate, while S-RVT demonstrates a 10.5 % improvement compared to RVT Goyal et al. (2023). For tasks demanding high-precision manipulation, such as *Insert Peg* and *Sort Shape*, our methods achieve success rates approximately 2 times of the state-of-the-art approaches.

The terms $\sigma_t$, $\sigma_r$, and $\sigma_s$ represent the learned task-dependent uncertainties. By optimizing this loss function, the model dynamically adjusts the weighting of each task's loss based on its uncertainty, effectively balancing the learning process across tasks. The logarithmic terms act as regularizers, preventing the uncertainties from becoming excessively large.

## 4 EXPERIMENT

We evaluate the performance of S-RVT and S-RVT2 through a series of comprehensive experiments conducted in both simulation and real world. This section is structured as follows: experimental setup, the results from simulation, ablations, and real-world experiments.

### 4.1 EXPERIMENTAL SETUP

In this study, we employ a widely-used benchmark for multi-task manipulation. This benchmark, originating from RLBench James et al. (2020), has been adopted by several related studies Shridhar et al. (2023); Goyal et al. (2023; 2024); Gervet et al. (2023). It encompasses 18 diverse tasks, ranging from non-prehensile actions like *push buttons* to common pick-and-place operations such as *place wine*, and high-precision tasks like *insert peg*. Each task is accompanied by a language description and consists of 2 to 60 variations, including handling objects of different colors or in various locations. The experiments use a Franka Panda robot equipped with a parallel jaw gripper. Both the task environment and the robot are simulated using CoppeliaSim Rohmer et al. (2013). The system processes RGB-D images at a resolution of $128 \times 128$, captured by four noiseless cameras positioned at the robot's front, left shoulder, right shoulder, and wrist. For training and evaluating the S-RVT model, we use the same dataset as employed in PerAct Shridhar et al. (2023) and RVT-2 Goyal et al. (2024) studies. Specifically, 100 demonstrations per task are used for training, while 25 previously unseen demonstrations / trajectories are for testing.

### 4.2 SIMULATION EXPERIMENT

We train S-RVT using a single node with 8 NVIDIA 3090 GPUs. The training process for S-RVT and S-RVT2 consists of 80K steps, employing a cosine learning rate decay schedule with an initial warmup period of 2K steps. We set the batch size to 256 ($32 \times 8$) and the initial learning rate to $1.8 \times 10^{-4}$. For evaluation, we use the final trained model. Given that RLBench James et al. (2020)

| Row ID | Method | SPR | HSP | Focal | Uncer. | # Views | SR # | Avg. Succ. | Diff. wrt. base |
|--------|--------|-----|-----|-------|--------|---------|------|------------|-----------------|
| 1  | S-RVT2 | ✓ | ✓ | ✓ | ✓ | 4 | 4 | 87.8 | 0     |
| 2  | S-RVT2 | ✗ | ✓ | ✓ | ✓ | 4 | 4 | 82.3 | - 5.5 |
| 3  | S-RVT2 | ✓ | ✗ | ✓ | ✓ | 4 | 4 | 86.9 | - 0.9 |
| 4  | S-RVT2 | ✓ | ✓ | ✗ | ✓ | 4 | 4 | 87.2 | - 0.6 |
| 5  | S-RVT2 | ✓ | ✓ | ✓ | ✗ | 4 | 4 | 86.1 | - 1.7 |
| 6  | S-RVT2 | ✓ | ✓ | ✓ | ✓ | 3 | 4 | 81.9 | -5.9  |
| 7  | S-RVT2 | ✓ | ✓ | ✓ | ✓ | 5 | 4 | 84.1 | -3.7  |
| 8  | S-RVT2 | ✓ | ✓ | ✓ | ✓ | 4 | 2 | 87.0 | -0.8  |
| 9  | S-RVT2 | ✓ | ✓ | ✓ | ✓ | 4 | 1 | 86.9 | -0.9  |
| 10 | S-RVT2 | ✗ | ✗ | ✗ | ✗ | 3 | 1 | 81.4 | -6.5  |
| Row ID | Method | SPR | HSP | Focal | Uncer. | # Views | SR # | Avg. Succ. | Diff. wrt. base |
| 11 | S-RVT | ✓ | ✓ | ✓ | ✓ | 4 | 4 | 73.4 | 0     |
| 12 | S-RVT | ✗ | ✓ | ✓ | ✓ | 4 | 4 | 70.0 | - 3.4 |
| 13 | S-RVT | ✓ | ✗ | ✓ | ✓ | 4 | 4 | 66.9 | - 6.5 |
| 14 | S-RVT | ✓ | ✓ | ✗ | ✓ | 4 | 4 | 68.1 | - 5.3 |
| 15 | S-RVT | ✓ | ✓ | ✓ | ✗ | 4 | 4 | 71.7 | - 1.7 |
| 16 | S-RVT | ✓ | ✓ | ✓ | ✓ | 3 | 4 | 69.2 | -4.2  |
| 17 | S-RVT | ✓ | ✓ | ✓ | ✓ | 5 | 4 | 71.4 | -2.0  |
| 18 | S-RVT | ✓ | ✓ | ✓ | ✓ | 4 | 2 | 67.1 | -6.3  |
| 19 | S-RVT | ✓ | ✓ | ✓ | ✓ | 4 | 1 | 63.5 | -9.9  |
| 20 | S-RVT | ✗ | ✗ | ✗ | ✗ | 3 | 1 | 60.2 | -13.2 |

Table 2: Ablations on S-RVT and S-RVT2. Starting from the third column, we investigate the impact of SPR, HSP, focal loss, uncertainty weighting, the number of virtual views, and the upscaling factor in super-resolution on model performance. We report the model's performance on average success rate across 18 simulation tasks and the difference with the base model.

employs a sampling-based motion planner, each model is evaluated 5 times on each task, and we report both the mean performance and variance.

We compare our S-RVT and S-RVT2 with various baselines. These include image-to-action behavioral cloning models, Image-BC (CNN) Jang et al. (2022); Shridhar et al. (2023) and Image-BC (ViT) Jang et al. (2022); Shridhar et al. (2023), utilizing CNN and ViT backbones respectively. Additionally, we compare against models specifically designed for 3D object manipulation, such as C2F-ARM-BC James et al. (2022), PerAct Shridhar et al. (2023), HiveFormer Guhur et al. (2023), RVT Goyal et al. (2023), PolarNet Chen et al. (2023b), Act3D Gervet et al. (2023) and RVT-2 Goyal et al. (2024). All baselines, S-RVT and S-RVT2 are trained and evaluated using $128 \times 128$ input images, whereas Act3D Gervet et al. (2023) employs $256 \times 256$ images.

Our S-RVT demonstrates significant improvements over RVT Goyal et al. (2023) across 18 tasks in RLBench, achieving a $10.5\%$ increase in average success rate. S-RVT2 outperforms RVT-2 Goyal et al. (2024), with the success rate rising from $81.4\%$ to $87.8\%$, a $6.4\%$, as illustrated in Table 1. For high-precision manipulation tasks, such as *Insert Peg*, S-RVT2 exhibits performance 2.3 times superior to the state-of-the-art. Similarly, in the *Sort Shape* task, S-RVT2 achieves a $2\times$ improvement.

### 4.3 ABLATION STUDY

We conduct comprehensive ablation study to demonstrate the effectiveness of each component in our method. The results are presented in Table 2. Row 1 and Row 11 in Table 2 represent the results with the standard setup of our S-RVT2 and S-RVT. We perform the following ablations for both S-RVT and S-RVT2:

- **SPR**: whether the Super Point Renderer is used. As shown in Row 2 and Row 12 of Table 2, removing SPR leads to performance drops of 5.5 % for S-RVT2 and 3.4 % for S-RVT. This indicates that SPR contributes to the model's ability to handle occlusions.

- **HSP**: whether the Hierarchical Sampling Policy is employed. From Row 3 and Row 13, we observe that without HSP, the performance decreases by 0.9 % for S-RVT2 and 6.5 % for S-RVT. The substantial drop in S-RVT suggests that HSP is crucial for accurate 3D pose estimation.

- **Focal**: whether focal loss is used. Comparing Row 4 and Row 14 to the base models, we find performance decreases of 0.6 % for S-RVT2 and 5.3 % for S-RVT when focal loss is not utilized. This demonstrates that focal loss effectively enhances super-resolution heatmap supervision.

- **Uncer.**: whether uncertainty weighting is employed to balance losses. The results in Row 5 and Row 15 show performance drops of 1.7 % for both S-RVT2 and S-RVT without uncertainty weighting. This suggests that uncertainty weighting helps in balancing losses.

- **# Views**: the quantity of virtual views utilized and test the following configurations:

  - **3 views** (*top, front, right*): As seen in Row 6 and Row 16, using only 3 views results in performance drops of 5.9 % for S-RVT2 and 4.2 % for S-RVT. Fewer views may lead to insufficient spatial information.
  - **4 views** (*top, front, right, down*): This configuration yields the best performance and is used in our base models.
  - **5 views** (*top, front, left, right, down*): As shown in Row 7 and Row 17, adding an extra view does not improve performance and leads to decreases of 3.7 % for S-RVT2 and 2.0 % for S-RVT. The additional left view may introduce redundancy or conflicting information, potentially overwhelming the model with unnecessary data.

  These results suggest that using 4 views provides the optimal balance between performance and computational cost.

- **SR #**: the upsampling ratio of the output heatmap compared to the input S-MVT virtual view by experimenting with:

  - **2× upsampling**: As seen in Row 8 and Row 18, reducing the upsampling ratio to 2× causes performance drops of 0.8 % for S-RVT2 and 6.3 % for S-RVT. This indicates that lower-resolution heatmaps may lead failure cases in high-precision tasks.
  - **1× upsampling**: In Row 9 and Row 19, 1× upsampling further decreases performance by 0.9 % for S-RVT2 and 9.9 % for S-RVT.

  This indicates that higher resolution heatmaps (4× upsampling) are beneficial for model performance, particularly for tasks requiring fine precision and detailed spatial understanding.

## 4.4 REAL-WORLD EXPERIMENT

In this subsection, we present our experiments in real world. We discuss four aspects: the experimental setup, dataset, training and evaluation details, and results.

**Experimental Setup.** Our experimental setup for real world comprises a $1400mm \times 700mm$ table, with a UFAC-TORY xArm v7 robotic arm and gripper fixed at the center of the table's long edge. An Intel RealSense L515 LiDAR camera is mounted at a fixed third-person viewpoint. We perform extrinsic calibration between the camera frame and the robot base frame to enable the transformation of RGB-D point cloud from the camera to the robot base. The S-RVT model processes RGB-D point cloud in the robot base frame and generates the key pose for the subsequent action, encompassing translation, rotation, and gripper state. Following each action execution, the robotic arm acquires a new RGB-D point cloud, and this process iterates until either task completion or reaching the maximum number of execution steps (25).

| Task | # of vari. | # of train | # of test | Succ. |
|------|------|------|------|------|
| Put item in drawer | 3 | 20 | 10 | 50 % |
| Stack blocks | 5 | 25 | 10 | 70 % |
| Place fruit on plate | 3 | 15 | 10 | 80 % |
| Plug charger | 1 | 15 | 10 | 60 % |
| All tasks | 12 | 90 | 40 | 65 % |

Table 3: Results of S-RVT. We conduct experiments on four tasks in real-world settings, recording the number of variations for each and the number of human demonstrations collected for training. We perform 10 test episodes for each task, reporting the success rate achieved in evaluation.

**Dataset.** We conduct experiments on the following tasks in real world: *put item in drawer, stack blocks, place fruit on plate, and plug charger*. These tasks encompass various challenges, including semantic understanding (*e.g.*, *place fruit on plate*), long-sequence decisions (*e.g.*, *put block in drawer* and *stack blocks*), and high-precision manipulations (*e.g.*, *plug charger*). The variations of each task are discussed in Appendix A.2. We collect 20, 15, 20, and 15 demonstrations for the tasks *put block in drawer, stack blocks, place fruit on plate, and plug charger*, respectively. The number

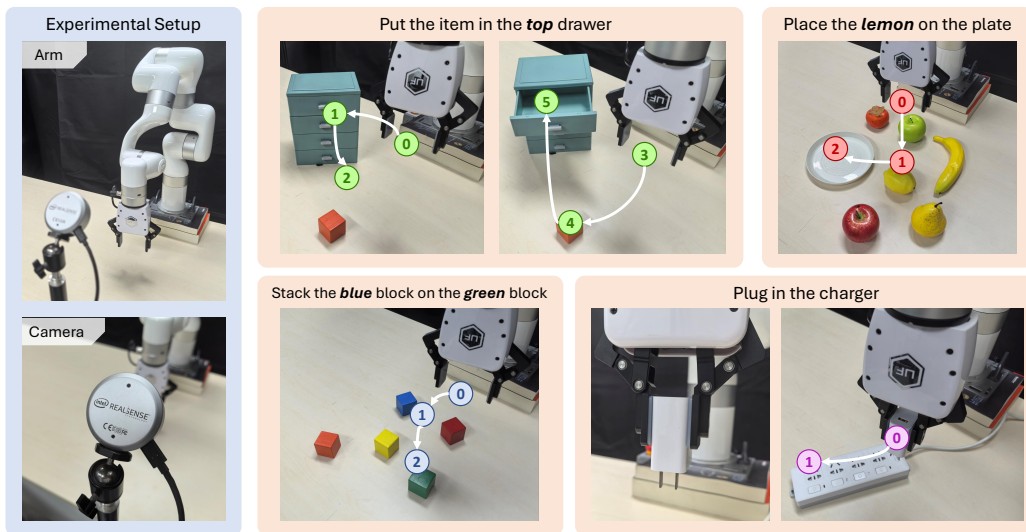

Figure 5: Illustration of the experimental setup and tasks in real world. The setup comprises a robotic manipulator and a third-person RGB-D camera. Four manipulation tasks are implemented: *put item in drawer, stack blocks, place fruit on plate, and plug charger*.

of demonstrations for each task is determined by its complexity and the degree of variability in task configurations. For instance, *put block in drawer* is a complex multi-step task, while *stack blocks* presents high variability due to the numerous possible color combinations of the blocks.

**Training and Evaluation Details.** A single model is trained for all four tasks, with training conducted over 100 epochs using a fixed learning rate of $5 \times 10^{-4}$. The final trained model is utilized for result evaluation. During testing, object positions and orientations are varied from those in the training data to assess generalization. A total of 10 test episodes are conducted. Task completion is defined as successful if achieved within 25 steps; exceeding this step limit or encountering collisions results in failure. Success signals are given by environment, for instance, in the *place fruit on plate* task, success is determined by the presence of the fruit within the plate.

**Results.** As shown in Table 3, despite using few number of human demonstrations, we still achieve a high success rate in real world. This demonstrates the generalization capability of our S-RVT in imitation learning. However, there are still some failure cases, which we attribute mainly to two factors. First, since we use only one third-person view camera, keypoints may become completely occluded during the experiment; for example, when the gripper is grasping an apple, it may entirely block the plate, causing the model to fail to predict the next key pose. Second, due to the limited number of demonstrations, the model's semantic understanding and discriminative ability are relative weak; for instance, the model may confuse similar objects like yellow pear and yellow lemon.

## 5 CONCLUSION

In this paper, we address the limitations of previous virtual view-based methods, focusing on occlusion problems and resolution constraints in the action space. To overcome these challenges, we introduce the Super Robot View Transformer (S-RVT). Our approach incorporates a Super Point Renderer that re-renders 3D point cloud into virtual images from fixed viewpoints, reducing information loss due to occlusion. The Super-resolution Multi View Transformer (S-MVT) then processes these images to generate high-resolution heatmaps representing 3D key pose translation probabilities on 2D planes. Finally, we employ a Hierarchical Sampling Policy (HSP) to sample and determine the 3D key pose. Our method demonstrates substantial improvements across 18 tasks, increasing performance from $81.4\%$ to $87.8\%$ compared to the previous state-of-the-art. Notably, for tasks requiring high-precision manipulation, our approach achieves a two-fold improvement in effectiveness. While current virtual view-based methods typically predict discrete key poses for the gripper, our future work will explore the potential for continuous trajectory prediction, further advancing the field of robotic manipulation.

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

## A APPENDIX

This section presents detailed information on the following aspects: 1) the model architecture of the Super-resolution Multi-View Transformer (S-MVT), 2) the real-world experiment, 3) data augmentation and 4) visualization of simulation results.

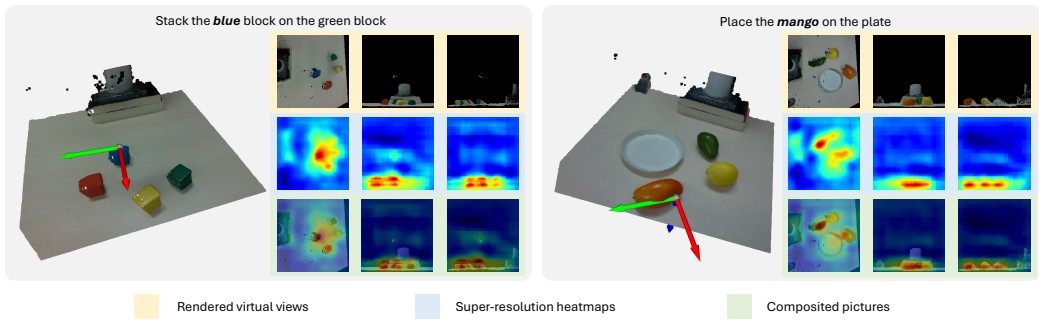

Figure 6: Model architecture of Super Multi-View Transformer (S-MVT). S-MVT takes multi-view virtual images and task descriptions as inputs. The multi-view virtual images undergo image patching and self-attention to extract image features. These image features are subsequently concatenated with language features derived from the task description. The concatenated features are then processed through another self-attention to generate heatmap features. The heatmap features are upsampled to produce super-resolution multi-view heatmaps. Leveraging the 3D translation predicted by the Hierarchical Sampling Policy (HSP), conditioned sampling is performed on the heatmap features. Finally, the sampled features are processed through two separate MLPs to estimate the rotation and gripper state.

Figure 7: Visualization of intermediate results in real-world tasks. We visualize two different tasks in real world: *stack the blue block on the green block* and *place the mango on the plate*. For the former task, the next key pose is to grasp the blue block, as predicted gripper coordinate indicates. For the latter task, the next key pose is to grasp the mango.

## A.1 MODEL ARCHITECTURE OF S-MVT

As illustrated in Figure 6, our Super-resolution Multi-View Transformer maps the input multi-view virtual images and task description to the translation, rotation, and gripper state of the next key pose. Initially, we patchify the multi-view virtual images and process them through four attention layers to obtain multi-view image features. We then concatenate these features with language features extracted by CLIP text encoder. The resulting combined feature vector is processed through another four attention layers to generate heatmap features. This heatmap features undergo super-resolution upsampling to produce multi-view super-resolution heatmaps.

To determine the rotation and gripper state of the next key pose, we employ conditional sampling. During the training phase, we project the ground-truth 3D translation onto the heatmap plane and select the heatmap feature corresponding to the projected 2D coordinates. This feature is then processed through two separate MLPs to output the rotation and gripper state, respectively. In the inference phase, we utilize the Hierarchical Sampling Policy (HSP) to predict the 3D translation, which serves as a condition for sampling the feature. This sampled feature is subsequently processed through MLPs to obtain the rotation and gripper state.

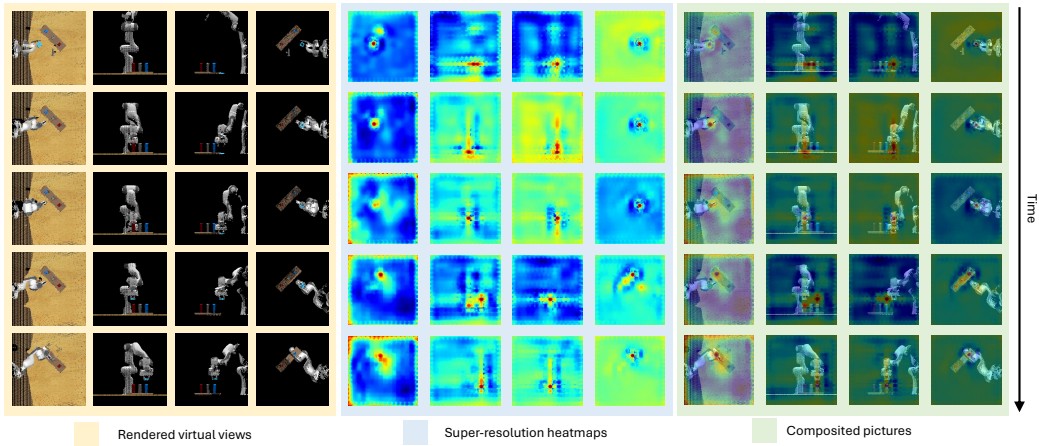

Figure 8: Visualization of intermediate results in simulation task *put the ring on the azure spoke*, including a series of key poses.

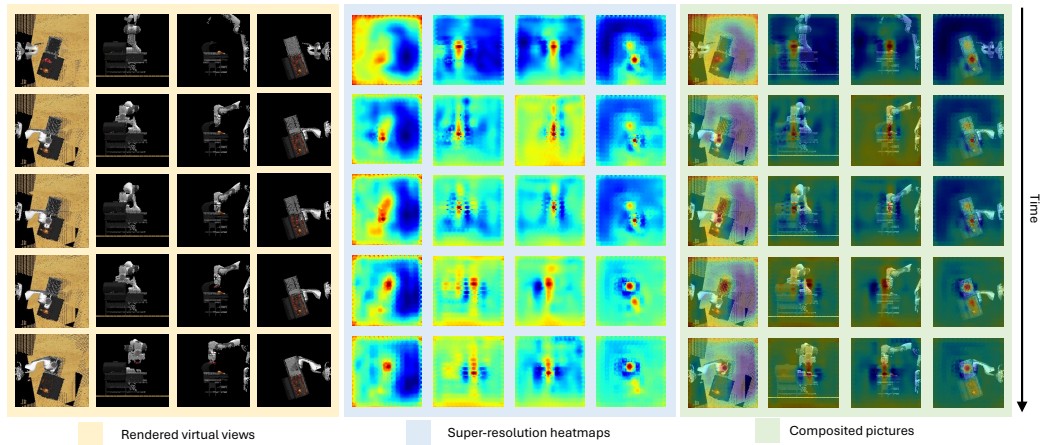

Figure 9: Visualization of intermediate results in simulation task *take the steak off the grill*, including a series of key poses.

### A.2 THE REAL-WORLD DATASET

The following are variations of different tasks in real world.

- Put the item in the # drawer: *top, middle, bottom*.

- Put the # on the plate: *lemon, mango, avocado*.

- Stack the # block on the # block: *red on blue, red on yellow, red on green, blue on yellow, blue on green, blue on orange, yellow on blue, yellow on green, yellow on orange, green on red, green on yellow, green on blue*.

- Plug in the charger.

We also show our intermediate results of real-world tasks: *stack the blue block on the green block* and *place the mango on the plate* in Figure 7. For observation in Figure 7, the next key pose is *grasp the blue block* and *grasp the mango* respectively. We visualize the rendered virtual views, heatmaps and their combinations. The top view of heatmap clearly shows the trained model focuses on the task relevant object and produce feasible output. We use three views: top, front, right in real-world settings for simplicity, while in simulation, we use four views for better performance.

| Task | # of vari. | # of train | # of test | Model RVT | RVT-2 | S-RVT | S-RVT2 |
|------|-----------|-----------|----------|-----------|-------|-------|--------|
| Put item in drawer | 3 | 20 | 10 | 30 % | 50 % | 50 % | 60 % |
| Stack blocks | 5 | 25 | 10 | 70 % | 70 % | 70 % | 80 % |
| Place fruit on plate | 3 | 15 | 10 | 70 % | 80 % | 80 % | 90 % |
| Plug charger | 1 | 15 | 10 | 10 % | 70 % | 60 % | 80 % |
| All tasks | 12 | 90 | 40 | 45 % | 67.5 % | 65 % | 77.5 % |

Table 4: Results of S-RVT and S-RVT2 compared to RVT Goyal et al. (2023) and RVT-2 Goyal et al. (2024). We conduct experiments on four tasks in real-world settings, recording the number of variations for each and the number of human demonstrations collected for training. We perform 10 test episodes for each task, reporting the success rate achieved in evaluation.

## A.3 DATA AUGMENTATION

We implement two types of data augmentation in both simulation and real world:

1. Point Cloud Augmentation: We apply random transformations to the input RGB point cloud to enhance the model's learning of interest point positions relative to language instructions and the rotation of subsequent key poses. These transformations include:
   - Random translation with a magnitude of 0.125 relative to the point cloud dimensions.
   - Random rotation around the z-axis up to 45 degrees.
2. RGB Image Augmentation: For the rendered RGB images, we introduce random perturbations to the pixel values. Specifically, we add a random noise term $\epsilon$ to each pixel value, where:

$$I_{augmented} = I_{original} + \epsilon, \quad \epsilon \sim \mathcal{U}(-0.05, 0.05). \tag{4}$$

Here, $I_{augmented}$ and $I_{original}$ represent the augmented and original pixel intensities, respectively, and $\epsilon$ is sampled from a uniform distribution between -0.05 and 0.05.

## A.4 VISUALIZATION OF SIMULATION RESULTS

We visualize the simulation results in Figure 8 and Figure 9. The two pictures show intermediate results of task *put the ring on the azure spoke* and *take the steak off the grill*. We also attach a video describing the robot executing task in simulation in the supplementary materials.

## A.5 REAL WORLD BASELINES

We compare our S-RVT and S-RVT2 with baseline methods of RVT Goyal et al. (2023) and RVT-2 Goyal et al. (2024), as illustrated in table 4.Our S-RVT achieves notable enhancement over RVT in high-precision manipulation tasks, such as the *plug charger* task, with success rate improvements of 50 %.

## A.6 DATA EFFICIENCY

Data scarcity and heterogeneity are major challenges in current manipulation tasks. To address heterogeneity, researchers have experimented with uniquely designed model architectures that train small-parameter networks for each specific embodiment while keeping the backbone network parameters fixed. This approach offers a pathway to mitigating data heterogeneity. Nevertheless, data scarcity is a more fundamental problem. Unlike the era of internet AI where large datasets are readily available online, robotics researchers cannot easily obtain vast amounts of data from the internet. Consequently, there is a growing interest in how to train robust and highly generalizable robots using limited data.

Existing methods like Diffusion Policy (DP) generate robust actions through denoising processes but require hundreds of human demonstrations to achieve convergence. Similarly, Action Chunking

with Transformers (ACT) collects human demonstrations through specialized mechanical setups, yet it still needs dozens of demonstrations to perform tasks robustly. In contrast, our proposed S-RVT framework addresses data efficiency by transforming raw observations into point cloud space and applying translation and rotation augmentations, significantly enhancing the diversity of virtual images. This allows our model to achieve convergence with as few as 10 demonstrations per task in real-world experiments. Moreover, the model learns correlations across multiple tasks, presenting possibilities for scaling up.

## A.7 FUTURE WORK

In future work, we aim to address camera occlusion in real-world scenarios from both temporal and spatial perspectives. Temporally, since objects are not continuously occluded throughout the sequence, enabling S-RVT to retain memory of past observations is valuable. Spatially, deploying multiple cameras from different viewpoints help mitigate occlusion. Regarding the limited number of demonstrations, generating additional demonstrations from synthetic data could enable the training of a robust robot using only a few real-world demos.

