# OpenReview forum: "Super Robot View Transformer"
_ICLR.cc/2025/Conference — Submitted to ICLR 2025_

### Official Review · Reviewer_Cpa1 · 2024-11-01

**Soundness:** 3
**Presentation:** 2
**Contribution:** 3
**Rating:** 6
**Confidence:** 3

**Summary:**

Building on prior work, RVT and RVT-2, this study introduces S-RVT to address limitations like occlusion issues and resolution constraints. Specifically, it presents S-PR (Super Point Render) to enhance rendering and reduce occlusion artifacts, S-MVT (Super-resolution Multi-View Transformer) to integrate super-resolution to output heatmaps, and HSP (Hierarchical Sampling Policy) for accurate 3D pose estimation through a coarse-to-fine sampling approach. Experimental results show that S-RVT outperforms previous methods.

**Strengths:**

* The approach of addressing limitations in prior work, such as handling occlusion issues in the rendering process and overcoming resolution constraints in pose prediction, is valuable.
* The authors perform extensive experiments, including comparisons with baseline models and ablation studies, to demonstrate the effectiveness of the proposed components.
* They also conduct several real-world experiments and provide the video evidence.

**Weaknesses:**

* It would be beneficial to clearly specify the differences between the proposed method and prior work, indicating which contributions are adopted from previous studies and which are newly introduced in this paper. For example, in Section 3.2, RVT-2 appears to have already implemented z-ordering and screen-space splatting techniques, yet this is not clarified here. Additionally, in Section 3.3, lines 255 to 263 (nearly half the paragraph) contain content similar to RVT and RVT-2, it would be helpful to focus more on the novel methods introduced in this work. Clearly distinguishing between techniques inherited from previous work and unique innovations would improve understanding and highlight the contributions of this study.
* In Section 3.3, additional details on the upsampling process would be helpful for clarity. Could the authors expand on how the upsampling is implemented?
* The authors are encouraged to provide further analysis of the experimental results:
  - In Table 1, S-RVT performs worse than RVT in tasks such as 'put in safe' and 'sweep to dustpan,' and S-RVT2 performs worse than RVT-2 in 'slide block' and 'sweep to dustpan.' Although these lower scores are acceptable for specific tasks, more in-depth analysis of why the proposed method underperforms in these cases would strengthen the findings.
  - In the ablation study, the impact of each component varies between S-RVT and S-RVT2. For instance, SPR is more critical for S-RVT2, whereas HSP has a greater influence on S-RVT. Additional analysis on why these components affect the models differently would offer valuable insights.
* The authors are also encouraged to discuss failure cases of the proposed method.

**Questions:**

* In line 257, the statement *"S-MVT generates heatmaps with $sr$ times higher resolution"* is unclear, as $sr$ is not introduced in the preceding paragraphs.
* In line 260, should this result in $16^2$ patches instead of $16$ patches?

---

> ### Author Response · Authors · 2024-11-21
> **To Reviewer Cpa1 1**
>
> a) **W1** It would be beneficial to clearly specify the differences between the proposed method and prior work, indicating which contributions are adopted from previous studies and which are newly introduced in this paper. For example, in Section 3.2, RVT-2 appears to have already implemented z-ordering and screen-space splatting techniques, yet this is not clarified here. Additionally, in Section 3.3, lines 255 to 263 (nearly half the paragraph) contain content similar to RVT and RVT-2, it would be helpful to focus more on the novel methods introduced in this work. Clearly distinguishing between techniques inherited from previous work and unique innovations would improve understanding and highlight the contributions of this study.
>
> **W2** In Section 3.3, additional details on the upsampling process would be helpful for clarity. Could the authors expand on how the upsampling is implemented?
>
> **Q1** In line 257, the statement *"S-MVT generates heatmaps with sr times higher resolution"* is unclear, as sr is not introduced in the preceding paragraphs.
>
> **Q2** In line 260, should this result in $16^2$ patches instead of 16 patches?
>
> Thank you for your valuable feedback and insightful comments. We agree that we should focus more on our own innovations rather than describing others' work. Our intention is to provide sufficient background for readers unfamiliar with RVT and RVT-2 to quickly understand our method. To address your concerns and clarify our contributions, we have revised the section on the Super-resolution Multi-View Transformer (S-MVT) as follows (this revision also addresses your points about the **introduction of the super-resolution factor *sr*** and **the** **details of the upsampling process**):
>
> *The task descriptions are processed through the CLIP (ResNet-50) text encoder to extract features, which, together with the rendered multi-view images, are then fed into the Super-resolution Multi-View Transformer (S-MVT). As shown in Figure 2, S-MVT generates super-resolution heatmaps; we denote the super-resolution factor as sr. Additionally, S-MVT outputs the rotation and gripper opening predictions for the next key pose. Specifically, the virtual images and language features are processed through an MVT structure similar to that in RVT, producing feature maps. These feature maps undergo upsampling to produce an sr-fold super-resolution heatmap, representing the probability distribution of possible 3D poses projected onto the 2D plane. Our upsampling employs an Efficient Up-convolution Block (EUCB), which uses Depthwise Separable Convolution (DWC) to reduce computational cost and parameter count while improving output resolution and preserving feature details. To predict the rotation and gripper state, we sample features from the image patch corresponding to the projected 3D position of the predicted next key pose on the virtual view. These sampled features are then processed through an MLP to estimate the rotation and gripper opening. This conditional sampling approach is employed because the gripper’s rotation and opening are intrinsically linked to its translation, thereby yielding more plausible predictions. The details of our model architecture are discussed in Appendix A.1.*
>
> For question regarding **S-PR**, we have updated the paragraph as follows:
>
> *Specifically, we first introduce an occlusion handling policy for the right and down views. We preprocess the point cloud using CUDA-accelerated DBSCAN clustering in the color space to filter out occluding elements like the tabletop while retaining task-relevant features. The robotic arm is retained as its pose provides valuable information about task progression. Second, we use an orthographic camera model to project the point cloud onto the image plane, preserving geometric relationships without perspective distortion. This rendering process comprises three key steps and is implemented in CUDA for acceleration which is introduced by Goyal et al. (2023). 1) The 3D points are projected onto the 2D image plane by converting them into image coordinates using GPU-accelerated matrix operations. 2) Z-ordering is applied to identify the point with the smallest depth for each pixel. 3) Screen-space splatting is used to model the points as finite-radius discs rather than singular pixels. As illustrated in Figure 3, in tasks such as put the ring on the azure spoke, standard point rendering fails to provide clear visual cues due to occlusions, making it difficult for the model to learn necessary rotations. Our S-PR mitigates this problem by multi-view rendering from different viewpoints, enhancing the model’s understanding in complex manipulations. In Figure 3, the filtered down view of our S-PR avoids the occlusions and explicitly shows the azure spoke and ring positions.*

---

> > ### Comment · Reviewer_Cpa1 · 2024-11-27
> >
> > Thank you for the detailed response and additional discussion. I don't have further questions, and I remain positive about this paper.

---

> > ### Comment · Reviewer_Cpa1 · 2024-11-27
> >
> > Thank you for the detailed response and additional discussion. I don't have further questions, and I remain positive about this paper.

---

> ### Author Response · Authors · 2024-11-21
> **To Reviewer Cpa1 2**
>
> b) **W3** The authors are encouraged to provide further analysis of the experimental results:
>
> - In Table 1, S-RVT performs worse than RVT in tasks such as 'put in safe' and 'sweep to dustpan,' and S-RVT2 performs worse than RVT-2 in 'slide block' and 'sweep to dustpan.' Although these lower scores are acceptable for specific tasks, more in-depth analysis of why the proposed method underperforms in these cases would strengthen the findings.
> - In the ablation study, the impact of each component varies between S-RVT and S-RVT2. For instance, SPR is more critical for S-RVT2, whereas HSP has a greater influence on S-RVT. Additional analysis on why these components affect the models differently would offer valuable insights.
>
> We appreciate your insightful comments. Regarding the tasks *put in safe* in S-RVT and *sweep to dustpan* in S-RVT2, the reductions in success rates compared to the original RVT and RVT2 are both less than 0.04. Given that each success rate is based on 25 test episodes, this difference corresponds to less than one successful attempt, which may indeed be attributed to random fluctuations.
>
> For the tasks *sweep to dustpan* in S-RVT and *slide block* in S-RVT2, we believe that the observed performance decrease may be due to an imbalance in learning across different tasks during training. We train the models for a total of 100 epochs. At 50 epochs, the success rates on these two tasks are higher than those of the baseline models. However, when training continue to 100 epochs, we observe that while the success rates for most tasks continue to surpass those of the baseline, the performance on these particular tasks begin to decline. We hypothesize that as the number of training epochs increases, the model may focus more on the more challenging tasks, leading to a potential "**forgetting**" of tasks that are learned well in the initial stages. This could result in an imbalance in the learning progress across different tasks. Since we report the success rates at 100 epochs, the performance decline on individual tasks like these may be reflected in the results.
>
> We appreciate the reviewer's insightful comments and agree that a more in-depth analysis of our ablation study results is necessary. Firstly, for **S-RVT2**, the **S-PR** is a more critical module. This is because, after the coarse-to-fine zoom-in process, the action space resolution in **S-RVT2** is sufficient to accomplish high-precision manipulation tasks. Therefore, **HSP** and **S-MVT** contribute less to **S-RVT2**'s performance. However, since **S-PR** addresses the occlusion issues in virtual views, it remains important for both **S-RVT** and **S-RVT2**. In contrast, the action space resolution in **S-RVT** is insufficient for high-precision tasks. This is why **HSP** and **S-MVT** play a dominant role in improving its performance. While our designed modules enhance precision, they also introduce some additional computational load. To further analyze the speed-accuracy trade-off, we have included additional experiments detailing the training time and inference speed, as shown in the table below:
>
> | Model  | Training Time (days) | Inference Speed (fps) | Avg. Success |
> | ------ | -------------------- | --------------------- | ------------ |
> | RVT    | 0.85                 | 20.9                  | 0.629        |
> | RVT2   | 0.92                 | 20.1                  | 0.814        |
> | S-RVT  | 0.88                 | 20.5                  | 0.734        |
> | S-RVT2 | 1.03                 | 19.2                  | 0.878        |
>
> As observed, there is a decrease in training time and inference speed compared to RVT and RVT-2. However, we believe that the improvements in accuracy provide greater value in terms of the speed-accuracy trade-off.

---

> ### Author Response · Authors · 2024-11-21
> **To Reviewer Cpa1 3**
>
> c) **W4** The authors are also encouraged to discuss failure cases of the proposed method.
>
> Thank you for your valuable insights. For the **failure analysis**, in fact, we have discussed failure cases in Section 4.4, analyzing **failures in real-world scenarios** from two aspects. First, because only a single third-person view camera is deployed in the real environment, task-related objects may become completely occluded during execution. Second, due to the limited number of human demonstrations, the learned features may not be sufficiently discriminative.
>
> For the **failure cases in simulation tasks**, we identify two main sources of failure:
>
> 1. **Long-horizon decision-making.** Because the RVT model does not incorporate past observations, it may struggle to accurately determine its current state and that of the environment, leading to failures in executing tasks that require memory of previous steps.
> 2. **Tasks requiring large-angle rotations and precise alignments** (*e.g.*, hanging a mug on a mug tree). For example, in the *Place Cups* task shown in Table 1, the difficulty arises from:
>    1. Predicting the correct rotation (aligning the mug's handle with the "branches" of the mug tree at an appropriate angle).
>    2. Ensuring that the robotic arm's trajectory, obtained through inverse kinematics (IK) to reach the predicted key pose of the end effector, does not collide with the mug tree.
>
> These challenges are limitations of our method, but they also highlight possible directions for future work. For better visualization, we provide some failure cases in [To Reviewer Cpa1](https://anonymous.4open.science/r/Super-Robot-View-Transformer-Rebuttal/README.md)

---

### Official Review · Reviewer_HUZo · 2024-11-02

**Soundness:** 2
**Presentation:** 1
**Contribution:** 2
**Rating:** 3
**Confidence:** 5

**Summary:**

This paper proposed a framework for multi-task imitation learning with key-state-based methods for robotic manipulation learning, especially for high-precision tasks. The author claims their model addresses the epistemic uncertainty of the proposed framework. The framework, SRVT, consists of three modules: the Super Point Renderer (S-PR), the Super-resolution Multi-View Transformer (S-MVT), and the Hierarchical Sampling Policy (HSP). This paper shows both simulation and real-world experiment results to evaluate the framework.

**Strengths:**

1. This paper conducts sufficient experiments to evaluate the proposed framework on various robotic manipulation tasks with different baseline methods.
2. The pictures in this paper are presented clearly.
3. Related works are comprehensively reviewed.

**Weaknesses:**

1. The overall writing and flow of the paper need considerable improvement. The abstract, introduction, and related works sections are repetitive and convey similar concepts. Moreover, a paper should streamline these sections to provide a progressive understanding.

2. This paper's primary claim that the proposed MVT module can advance epistemic uncertainty is not validated. To address such uncertainty, the paper must provide theoretical proofs, uncertainty analysis, and ablation studies. Some common methods, like conditional variable at risk, can be used for this.

**Questions:**

1. In the paper: "The 3D points are projected onto the 2D image plane by converting them into image coordinates using GPU accelerated matrix operations." Could the author give more details on the implementation and visualization of this?

1. Why the MVT module can address the epistemic uncertainty? Please provide detailed information.

2. In the HSP section, can the author provide some details about how HSP can solve the GPU memory overflow problem?

---

> ### Author Response · Authors · 2024-11-21
> **To Reviewer HUZo**
>
> a) **W2** This paper's primary claim that the proposed MVT module can advance epistemic uncertainty is not validated. To address such uncertainty, the paper must provide theoretical proofs, uncertainty analysis, and ablation studies. Some common methods, like conditional variable at risk, can be used for this.
>
> **Q2** Why the MVT module can address the epistemic uncertainty? Please provide detailed information.
>
> Thank you for your valuable feedback regarding the treatment of **uncertainty** in our paper. We appreciate your suggestion to provide additional experimental analysis on uncertainty, and we acknowledge the importance of rigorously addressing this aspect. However, we find it challenging to reflect the types of uncertainty discussed in our method through quantitative analysis. In our manuscript, the two types of uncertainty—*epistemic* and *aleatoric*—are introduced from an empirical perspective to highlight the issues present in existing methods and to motivate our proposed solutions. We have demonstrated, through comparisons with baseline methods and extensive ablation studies, that the performance of our model decreases without the incorporation of our proposed modules.
>
> For example, in task *put the ring on the spoke*, when robot executes the key pose of align the ring with the spoke, the ring in the top view will undoubtedly be occluded by the robotic arm. Thus, this observation introduces *epistemic uncertainty*. Additionally, to precisely align the ring with the spoke, the resolution of robotic action space should be fine enough. Therefore, the discretized action space introduces *aleatoric uncertainty*. From this perspective, S-PR is designed to tackle *epistemic uncertainty*, while S-MVT and HSP is for *aleatoric uncertainty*. From the results in Table 1 of the manuscript, we observe in task *Insert Peg* or *Sort Shape*, our success rates are 0.86 (vs. o.40) and 0.71 (vs. 0.36), which validates our motivation.
>
> Regarding the conditional value at risk (CVaR) metric you mention, we understand that it is used to measure risk within a task. However, quantifying such risk in our desktop manipulation tasks is not straightforward, as these tasks do not inherently involve considerations of risk in the traditional sense. Nevertheless, we will consider elaborating on this point in the revised manuscript to clarify our position.
>
> b) **Q1** In the paper: "The 3D points are projected onto the 2D image plane by converting them into image coordinates using GPU accelerated matrix operations." Could the author give more details on the implementation and visualization of this?
>
> We appreciate the reviewer's question regarding the GPU-accelerated projection implementation. For complete transparency, we have made the detailed CUDA implementation available in the *render* subfolder at [To Reviewer HUZo](https://anonymous.4open.science/r/Super-Robot-View-Transformer-Rebuttal/README.md).
>
> c) **Q3** In the HSP section, can the author provide some details about how HSP can solve the GPU memory overflow problem?
>
> Thank you for raising this question about GPU memory management in HSP. As detailed in Section 3.4 of our manuscript below:
>
> *However, for our high-resolution heatmaps, uniformly distributing particles in 3D space at increasing resolutions leads to higher particle density, potentially causing GPU memory overflow. To address this issue, we develop the Hierarchical Sampling Policy. First, we sample at a lower resolution to obtain a coarse predicted pose. Subsequently, we perform a higher-density sampling in the vicinity of this initial prediction to refine the pose estimate, as depicted in Figure 4.*
>
> Additionally, we have performed a semi-quantitative analysis of GPU memory usage to illustrate how HSP mitigates memory issues. Assuming the resolution of the output action space is increased fourfold through super-resolution, i.e., $224 \times 4 = 896$, the traditional sampling method used in RVT-2 would require sampling $896^3$ points, which is approximately 0.7 billion points. In contrast, our HSP method samples only $224^3 \times 2$ points, amounting to approximately 0.022 billion points. By employing an even coarser sampling resolution during the initial sampling stage, the GPU memory consumption can be reduced further.

---

> > ### Comment · Reviewer_HUZo · 2024-11-27
> >
> > a) Thanks for the authors' feedback; I would like to argue that only from an empirical perspective to show the proposed method can address epistemic and aleatoric uncertainty does not establish soundness. Please show numerical results from an uncertainty analysis perspective. Some methods include heteroscedastic regression to disentangle epistemic uncertainty from aleatoric (data-related) uncertainty.
> >
> > b) Thanks for providing the code
> >
> > c) Thanks for the explanation.
> >
> > Overall, the authors provide a concrete rebuttal with clarifications on the technical aspects. However, the paper still fails to address the theoretical concerns.

---

### Official Review · Reviewer_Fpi1 · 2024-11-03

**Soundness:** 4
**Presentation:** 3
**Contribution:** 1
**Rating:** 5
**Confidence:** 4

**Summary:**

This paper presents Super Robot View Transformer (S-RVT) -- a series of techniques to improve Robot View Transformer.  It consists of 3 modules: the Super Point Renderer that mitigates occlusion artifacts, the Super-resolution Multi-View Transformer that performs superresolution to the output heatmaps, and the Hierarchical Sampling Policy that efficiently samples multi-view heatmaps in 3D space. The experiments suggest S-RVT obtains a consistent performance boost against RVT on the RLBench benchmark.

**Strengths:**

- This paper is well-written and easy to follow.
- The proposed modifications on RVT are intuitive, and the RLbench experiments verify the effectiveness of S-MVT in simulator settings.

**Weaknesses:**

- I doubt the fundamental rationality of the Super Point Renderer (S-PR) and Super-resolution Multi-View Transformer (S-MVT) modules.
  - According to my understanding, S-PR renders the objects occluded by the robot. It would definitely be effective in the simulator, but how would that be made possible in the real world?
  - Meanwhile, S-MVT aims to perform super-resolution to the multi-view images. However, why not just enhance the resolution of RGB-D images in the beginning? D515 could capture depth photos in a resolution of up to 1024x768, but the RGB-D images used in the paper only have a resolution of 128x128.
- While S-MVT is compared with MVT in the simulator setting, it is not compared with MVT in the real-world setting. I am particularly confused about how S-PR is implemented in real-world scenarios.

**Questions:**

- How is S-PR implemented in real-world scenarios?
- What will happen if the resolution of input images is raised?
- What's the performance of MVT in the real-world experiments?

I will consider raising scores if my concerns are addressed in the rebuttal period.

---

> ### Author Response · Authors · 2024-11-21
> **To Reviewer Fpi1 1**
>
> a) **W1.1** I doubt the fundamental rationality of the Super Point Renderer (S-PR) and Super-resolution Multi-View Transformer (S-MVT) modules.
>
> - According to my understanding, S-PR renders the objects occluded by the robot. It would definitely be effective in the simulator, but how would that be made possible in the real world?
>
> **Q1** How is S-PR implemented in real-world scenarios?
>
> We appreciate your valuable comments regarding **the effectiveness of the Super Point Renderer (S-PR) in real-world scenarios**. We address the occlusion problem by categorizing it into two types: 1) *occlusions caused by the **real** cameras' fields of view being obstructed*, and 2) *occlusions occurring in the **virtual** cameras' viewpoints*. For the first type, this effect can be easily mitigated by deploying multiple **real** cameras from different perspectives, thereby maximizing scene coverage and minimizing blind spots. For the second type, our S-PR module is specifically designed to reduce the impact of occlusions within the virtual camera views.
>
> In both real-world and simulated environments, our system takes as input RGB-D images captured by **real** cameras. In simulation, we maintain identical camera configurations as RVT-2, using the same number of cameras, identical mounting positions, and collecting the same number of images to ensure fair comparison. These images are transformed into point clouds in the robot base frame using corresponding extrinsics. From this point cloud, we generate multi-view observations by rendering from various **virtual** camera poses. It is important to note that if the objects are entirely occluded in all **real** camera views, it would indeed be impossible for the subsequent **virtual** cameras to recover the model's points of interest on the objects. However, such scenarios are rare. The utilization of multiple cameras (*e.g.*, left shoulder, right shoulder, wrist cameras) typically ensures comprehensive coverage of the scene. In our real-world experiments, we employ only a single third-person RGB-D camera, which is sufficient to observe the objects' points of interest. For further illustration, we have provided in the [To Reviewer Fpi1](https://anonymous.4open.science/r/Super-Robot-View-Transformer-Rebuttal/README.md) folder the following materials from our real-world experiments: 1) original images from the real camera, 2) S-PR rendered images and 3) the real experimental setup including robotic arm and the third-person camera.
>
> These examples demonstrate the effectiveness of S-PR in handling occlusions and rendering the necessary information for manipulation tasks in real-world settings. For further analysis, we conduct an experiment where we directly use images from real cameras as inputs to the MVT and compare the results with those obtained using virtual images from point cloud projections. A key advantage of using virtual projections is that they enable robust data augmentation through translation and rotation transformations, while such augmentation cannot be applied to real camera inputs without introducing point renderer. The augmentation details are presented in our supplementary materials.  The results are as follows:
>
> | Model        | Camera Location | Augmentation | Average Success |
> | ------------ | --------------- | ------------ | --------------- |
> | RVT          | Virtual         | $\checkmark$ | 0.629           |
> | RVT          | Real            | $\times$     | 0.229           |
> | S-RVT (ours) | Virtual         | $\checkmark$ | 0.734           |
> | S-RVT (ours) | Real            | $\times$     | 0.271           |
>
> From these results, it is evident that point cloud projection is indeed a core step in our framework. Using original images without point cloud projection leads to a significant performance degradation.

---

> ### Author Response · Authors · 2024-11-21
> **To Reviewer Fpi1 2**
>
> b) **W1.2** Meanwhile, S-MVT aims to perform super-resolution to the multi-view images. However, why not just enhance the resolution of RGB-D images in the beginning? D515 could capture depth photos in a resolution of up to 1024x768, but the RGB-D images used in the paper only have a resolution of 128x128.
>
> **Q2** What will happen if the resolution of input images is raised?
>
> Thank you for your insightful question regarding **the input image resolution**. Increasing image resolution can indeed be approached in two ways: 1) enhancing the resolution of the **real** camera images, and 2) enhancing the resolution of the **virtual** camera images. You are referring to the former. For the **real** camera images, as long as the resolution is sufficient to recognize the objects, it maintains the model's performance. Regarding the **virtual** camera images, simply increasing their resolution does not necessarily improve the model's learning because: 1) the existing resolution of the **virtual** camera images is already adequate for observation needs (please refer to the virtual images in the [To Reviewer Fpi1](https://anonymous.4open.science/r/Super-Robot-View-Transformer-Rebuttal/README.md) folder), and 2) for high-precision manipulation tasks, the model is constrained by the resolution of the output heatmaps, which corresponds to the quantization precision of the action space. If the resolution of the action space does not meet the task requirements, the results will not be optimal, regardless of the model's prediction accuracy.
>
> To validate this point, we conduct experiments in a real-world setting. We do not perform this experiment in a simulation environment because, to ensure a fair comparison, we must use the same resolution as the baseline method. To better assess the impact of changing the resolution of the **real**, **virtual**  or **heatmap** resolution on the model's performance, we compare the model's performance under different combinations of resolutions. The experimental setup and data are consistent with those in the main paper. We obtain different real camera resolutions by downsampling the original high-resolution images. The results are as follows:
>
> | Real Image Resolution | Virtual Image Resolution | Heatmap Resolution | Average Success |
> | --------------------- | ------------------------ | ------------------ | --------------- |
> | 960 $\times$ 540      | 224 $\times$ 224         | 896 $\times$ 896   | 65 %            |
> | 960 $\times$ 540      | 448 $\times$ 448         | 896 $\times$ 896   | 67.5 %          |
> | 960 $\times$ 540      | 112 $\times$ 112         | 896 $\times$ 896   | 57.5 %          |
> | 480 $\times$ 270      | 224 $\times$ 224         | 896 $\times$ 896   | 60 %            |
> | 960 $\times$ 540      | 224 $\times$ 224         | 448 $\times$ 448   | 52.5 %          |
> | 960 $\times$ 540      | 224 $\times$ 224         | 224 $\times$ 224   | 37.5 %          |
>
> The experimental results show that when we keep the **Real Image Resolution** unchanged and vary the **Virtual Image Resolution** (the first three rows), the success does not change significantly. Similarly, when the **Virtual Image Resolution** is kept constant and the **Real Image Resolution** is altered (the first and fourth rows), there is also no significant variation in success rate. However, when we reduce the **Heatmap Resolution** to a certain extent (the last row), there is a sharp decline in success rate. This indicates that, as long as the scene details are visible, the resolution of the action space is a more critical factor in the model's performance.
>
> c) **W2** While S-MVT is compared with MVT in the simulator setting, it is not compared with MVT in the real-world setting. I am particularly confused about how S-PR is implemented in real-world scenarios.
>
> **Q3** What's the performance of MVT in the real-world experiments?
>
> Thank you for your question regarding to **the performance comparison with baseline methods in real-world experiments**. To address this, we perform additional real-world experiments using the original RVT/RVT2 models as baselines. The results are as follows:
>
> | Task                 | RVT  | RVT-2  | S-RVT(ours) | S-RVT2(ours) |
> | -------------------- | ---- | ------ | ----------- | ------------ |
> | Put item in drawer   | 30 % | 50 %   | 50 %        | **60 %**     |
> | Stack blocks         | 70 % | 70 %   | 70 %        | **80 %**     |
> | Place fruit on plate | 70 % | 80 %   | 80 %        | **90 %**     |
> | Plug charger         | 10 % | 70 %   | 60 %        | **80 %**     |
> | All tasks            | 45 % | 67.5 % | 65 %        | **77.5 %**   |
>
> From the results above, our S-RVT achieves notable enhancement over RVT in high-precision manipulation tasks, such as the *plug charger* task, with success rate improvements of 50%. We have updated the results in the supplementary materials.

---

> > ### Comment · Reviewer_Fpi1 · 2024-11-26
> >
> > In the rebuttal period, the authors provide more empirical results to support the performance of S-RVT. However, my concerns are still not fully addressed because I am unable to obtain much insight beyond the empirical results. As a response, I decide to raise my score from 3 to 5.

---

### Official Review · Reviewer_4Vqv · 2024-11-03

**Soundness:** 3
**Presentation:** 3
**Contribution:** 2
**Rating:** 5
**Confidence:** 4

**Summary:**

The paper addresses the limitations in previous virtual view-based methods, focusing on the occlusion problems and the resolution constraints. To resolve these problems, it proposes the Super Robot View Transformer (S-RVT) comprising of three modules: the Super Point Renderer (S-PR) that enhances the rendering process to mitigate occlusion artifacts, the Super-resolution Multi-View Transformer (S-MVT) that integrates superresolution
to the output heatmaps, and the Hierarchical Sampling Policy (HSP) that samples multi-view heatmaps in 3D space to obtain accurate 3D poses. Experiments show that the proposed framework improves the performances of RVT and RVT2 in various setups.

**Strengths:**

- The authors have conducted abundant experiments and ablation studies in both sim and real.
- The experimental results look good. The proposed framework brings consistent improvements to RVT and RVT2 across different scenarios.
- The paper is well-written. The concepts and intuitions are explained together with concrete examples, making it very easy to understand.

**Weaknesses:**

- To my understanding, the paper is mainly addressing the uncertainty problems (in RVT or RVT-like robot learning frameworks): the aleatoric uncertainty is addressed by the virtual-view pointcloud rendering, and the epistemic uncertainty is addressed by the feature map superresolution. On one side, I like the intuitions discussed in the paper, on the other side, simply looking at the framework, the ways of resolving these problems look very straightforward, with a sequence of concrete engineering efforts. It would be good to have more concrete discussions, based on the method, on how RVT didn't address these uncertainties well and how the framework resolves these issues -- this can show better linkage between the high-level intuitions of the paper and the concrete steps in the method.
- A concrete question following the previous question is about the pointcloud rendering: It is simply done by a 2D projection, but what is the quality of the projected virtual views? Does it have any requirements on the placement of the (real) camera? Specifically, in the ablation study, it shows that going from 4 virtual views to 5 views decreases performance. I think this implies that the virtual views are not all of good quality which helps the algorithm to figure out better policies.

**Questions:**

- It would be nice if the authors could have some discussions on the data efficiency of the proposed framework. As discussed in the Related Work section, both transformer deployment and imitation learning with high precision require substantial training data. In this paper, the sim experiments use 100 demonstrations per task, and the real experiments use 15-20 demonstrations per task. How does it compare to other methods?
- Another very relevant question: For the real experiments, the paper states that "the number of demonstrations for each task is determined by its complexity and the degree of variability in task configurations." How much will this affect the performances? Specifically, for the two tasks "stack blocks" and "plug charger", how would the model perform if there are only 15 demonstrations, as in the other two tasks?
- I am curious, in Table 1 task Sweep to Dustpan, why is S-RVT success rate lower than RVT? Are there any specific features of this task that make it different from the others, or is it just a normal fluctuation in the measurement? (This is really just my curiosity. The overall experimental results look good to me, and a 10% success rate drop out of 25 tests here is acceptable.)

---

> ### Author Response · Authors · 2024-11-21
> **To Reviewer 4Vqv 1**
>
> a) **W1** To my understanding, the paper is mainly addressing the uncertainty problems (in RVT or RVT-like robot learning frameworks): the aleatoric uncertainty is addressed by the virtual-view pointcloud rendering, and the epistemic uncertainty is addressed by the feature map superresolution. On one side, I like the intuitions discussed in the paper, on the other side, simply looking at the framework, the ways of resolving these problems look very straightforward, with a sequence of concrete engineering efforts. It would be good to have more concrete discussions, based on the method, on how RVT didn't address these uncertainties well and how the framework resolves these issues -- this can show better linkage between the high-level intuitions of the paper and the concrete steps in the method.
>
> Thank you for your insightful comments. We agree that it is important to better highlight in our manuscript how our framework addresses the uncertainties that the original RVT did not resolve, and to clearly link the high-level intuitions with the concrete steps in our method. In response, we have revised the paragraph in the introduction regarding uncertainties to provide more concrete discussions. The modified content is as follows:
>
> *We advance beyond previous methods by addressing uncertainties inherent in both the model and the data (Kendall & Gal, 2017; Kendall et al., 2018). These uncertainties are broadly categorized into epistemic uncertainty, arising from limitations in the model, and aleatoric uncertainty, stemming from inherent variability in the data. For instance, in the task of placing the ring on the azure spoke, the former uncertainty refers to the possibility that the robot may misjudge the exact position of the spoke due to insufficient or biased training data. The latter uncertainty indicates that the robot fails to predict the next key pose due to occlusions. The original RVT framework discretizes the action space; however, such a coarse-grained action space is insufficient for accomplishing high-precision manipulation tasks, which contributes to epistemic uncertainty. Additionally, the points of interest on the manipulated objects are often occluded by the robot arm, making it difficult for the model to infer the next key pose based on the current observation, thus increasing aleatoric uncertainty. To address these uncertainties, we introduce the Super Robot View Transformer (S-RVT), a multi-task framework designed for high-precision manipulation tasks.*
>
> *Our S-RVT framework comprises three key modules: the Super Point Renderer (S-PR), the Super-resolution Multi-View Transformer (S-MVT), and the Hierarchical Sampling Policy (HSP). The S-PR mitigates aleatoric uncertainty by addressing observational uncertainties, particularly catastrophic occlusion, where critical visual obstructions impede task completion. The S-MVT and HSP work together to reduce epistemic uncertainty: S-MVT enhances model expressivity by generating super-resolution heatmaps with strong supervision, while HSP samples multi-view heatmaps in 3D space to obtain accurate 3D poses. By integrating these modules, S-RVT effectively reduces both epistemic and aleatoric uncertainties, advancing the state-of-the-art in high-precision robotic manipulation tasks. Notably, our method is a general boosting framework for virtual view-based approaches. Thus, we integrate it with both the RVT Goyal et al. (2023) and RVT-2 Goyal et al. (2024), yielding promising results across 18 challenging tasks in the RLBench benchmark James et al. (2020). For RVT, our S-RVT improves the average success from 0.629 to 0.734. Furthermore, our S-RVT2 achieves a success rate of 0.878, surpassing the state-of-the-art 0.814. In tasks requiring  high-precision, such as peg insertion, we establish a remarkable success rate of 0.86, achieving a relative 115 % improvement over the state-of-the-art performance of 0.40. We also demonstrate our method’s effectiveness in real world, as illustrated in Figure 1.*

---

> ### Author Response · Authors · 2024-11-21
> **To Reviewer 4Vqv 2**
>
> b) **W2** A concrete question following the previous question is about the pointcloud rendering: It is simply done by a 2D projection, but what is the quality of the projected virtual views? Does it have any requirements on the placement of the (real) camera? Specifically, in the ablation study, it shows that going from 4 virtual views to 5 views decreases performance. I think this implies that the virtual views are not all of good quality which helps the algorithm to figure out better policies.
>
> Thank you for your thoughtful questions and valuable feedback regarding the **rendering quality**. First, concerning your question about **the quality of point cloud projections**, we acknowledge that simply projecting point clouds does not achieve the high rendering quality of models like 3DGS, which possess strong rendering capabilities. However, we believe that rendering quality is not the critical factor in the model's ability to correctly output the key poses. As illustrated in the Figure 7 in our manuscript, as long as the interesting points of the object to be manipulated (*e.g.*, the contour of the banana in the banana-grasping task) are clearly presented, the model can make accurate predictions.
>
> To further substantiate this point, we conduct an experiment where we directly use images from real cameras as inputs to the MVT and compare the results with those obtained using virtual images from point cloud projections. A key advantage of using virtual projections is that they enable robust data augmentation through translation and rotation transformations, while such augmentation cannot be applied to real camera inputs without introducing point renderer. The augmentation details are presented in our supplementary materials.  The results are as follows:
>
> | Model        | Camera Location | Augmentation | Average Success |
> | ------------ | --------------- | ------------ | --------------- |
> | RVT          | Virtual         | $\checkmark$ | 0.629           |
> | RVT          | Real            | $\times$     | 0.229           |
> | S-RVT (ours) | Virtual         | $\checkmark$ | 0.734           |
> | S-RVT (ours) | Real            | $\times$     | 0.271           |
>
> From these results, it is evident that point cloud projection is indeed a core step in our framework. Using original images without point cloud projection leads to a significant performance degradation.
>
> Second, regarding your question about **the placement of the real cameras**, we agree that there are indeed considerations. The real cameras need to ensure that the objects to be manipulated in the scene are not occluded, which can be achieved by deploying multiple cameras from third-person viewpoints.
>
> In response to your observation that increasing the number of virtual views from 4 to 5 decreases performance, we have examined this issue closely. We invite you to refer to our response **d) To Reviewer jBkg**, where we discuss this in detail. Specifically, by replacing the right view with a left view when using 4 virtual views, we verify whether there is a difference between the right and left views. Our findings suggest that adding an extra left view may introduce conflicting and redundant information, which could explain the performance decline.

---

> ### Author Response · Authors · 2024-11-21
> **To Reviewer 4Vqv 3**
>
> c) **Q1** It would be nice if the authors could have some discussions on the data efficiency of the proposed framework. As discussed in the Related Work section, both transformer deployment and imitation learning with high precision require substantial training data. In this paper, the sim experiments use 100 demonstrations per task, and the real experiments use 15-20 demonstrations per task. How does it compare to other methods?
>
> Thank you for your valuable feedback. We appreciate your suggestion to discuss the data efficiency of our proposed framework, and we will incorporate the following discussion into the supplementary materials due to space limitations.
>
> *Data scarcity and heterogeneity are major challenges in current manipulation tasks. To address heterogeneity, researchers have experimented with uniquely designed model architectures that train small-parameter networks for each specific embodiment while keeping the backbone network parameters fixed. This approach offers a pathway to mitigating data heterogeneity. Nevertheless, data scarcity is a more fundamental problem. Unlike the era of internet AI where large datasets are readily available online, robotics researchers cannot easily obtain vast amounts of data from the internet. Consequently, there is a growing interest in how to train robust and highly generalizable robots using limited data.*
>
> *Existing methods like Diffusion Policy (DP) generate robust actions through denoising processes but require hundreds of human demonstrations to achieve convergence. Similarly, Action Chunking with Transformers (ACT) collects human demonstrations through specialized mechanical setups, yet it still needs dozens of demonstrations to perform tasks robustly. In contrast, our proposed S-RVT framework addresses data efficiency by transforming raw observations into point cloud space and applying translation and rotation augmentations, significantly enhancing the diversity of virtual images. This allows our model to achieve convergence with as few as 10 demonstrations per task in real-world experiments. Moreover, the model learns correlations across multiple tasks, presenting possibilities for scaling up.*
>
> d) **Q2** Another very relevant question: For the real experiments, the paper states that "the number of demonstrations for each task is determined by its complexity and the degree of variability in task configurations." How much will this affect the performances? Specifically, for the two tasks "stack blocks" and "plug charger", how would the model perform if there are only 15 demonstrations, as in the other two tasks?
>
> Thank you for raising this important question. We should first clarify that in our original experiments, *put item in drawer* and *stack blocks* have more demonstrations, while *plug charger* has 15 demonstrations. To address your concerns, we have extended our experiments by adding the results after reducing the number of demonstrations for the tasks *put item in drawer* and *stack blocks*, as shown below:
>
> | Task                 | # of variations | # of train | # of test | Success |
> | -------------------- | --------------- | ---------- | --------- | ------- |
> | *Put item in drawer* | 3               | 20         | 10        | 50 %    |
> | *Put item in drawer* | 3               | 15         | 10        | 40 %    |
> | *Stack blocks*       | 5               | 25         | 10        | 70 %    |
> | *Stack blocks*       | 5               | 15         | 10        | 40 %    |
>
> For the *put item in drawer* task, reducing the number of demonstrations leads to a slight decline in the model's performance. In contrast, for the *stack blocks* task, the success rate drops significantly to 40 %. This is because this task involves many variations (different combinations of block colors), and the limited number of demonstrations is insufficient to train discriminative visual features. Additionally, we speculate that inadequate lighting conditions in real world, resulting in darker scenes, may have reduced the quality of the collected visual data.
>
> e) **Q3** I am curious, in Table 1 task Sweep to Dustpan, why is S-RVT success rate lower than RVT? Are there any specific features of this task that make it different from the others, or is it just a normal fluctuation in the measurement? (This is really just my curiosity. The overall experimental results look good to me, and a 10% success rate drop out of 25 tests here is acceptable.)
>
> Thank you for your insightful question. We think it is just a normal fluctuation. Because in CoppelaSim, after providing the next key pose, the simulator determines the subsequent joint angles by sampling through inverse kinematics (IK). For tasks like *put item in drawer*, this sampling introduces randomness that can lead to unintended collisions, potentially causing an episode to fail. Therefore, we consider these performance differences to be within an acceptable range.

---

> ### Comment · Reviewer_4Vqv · 2024-12-03
>
> I want to thank the authors for their additional experiments and detailed explanations. My overall opinion on this paper is actually in-between borderline accept and borderline reject and it's a bit hard for me to decide which way to go. After reading the authors' responses as well as other discussions, I think my concerns are partially resolved. At this point, I may lean towards borderline reject, as my overall feeling is that the framework is quite unstable over the small design choices and hyperparameters.
>
> Specifically, my remaining concern may be in the performance drop w.r.t. more views -- the authors' explanations are that it could be due to "an extra left view may introduce conflicting and redundant information", but on the other hand, I think this explanation is not very coherent with other two points: (i) the multi-view transformer is designed for resolving ambiguity -- based on my understanding -- somehow leveraging the information from different views; (ii) the real cameras need to ensure that the objects to be manipulated in the scene are not occluded, in other words, the real cameras are trying to acquire more diverse information, but it sounds like the virtual ones should somehow avoid having too diverse information.

---

### Official Review · Reviewer_jBkg · 2024-11-04

**Soundness:** 3
**Presentation:** 2
**Contribution:** 3
**Rating:** 6
**Confidence:** 4

**Summary:**

This paper proposes an improvement over the RVT/RVT2 method by introducing enhancements such as S-PR, S-MVT, and HSP. The updates effectively address the problem of view occlusion in RVT from certain angles, particularly the top view. Extensive experiments demonstrate impressive performance gains with these adjustments, validating their effectiveness.

**Strengths:**

1. **Performance Improvement**: The proposed changes significantly enhance performance by mitigating occlusions from specific viewpoints, improving view flexibility.
2. **Robust Experimentation**: The paper includes multiple experiments to verify the efficacy of each introduced method, which supports the validity of the approach.

**Weaknesses:**

1. **Lack of Visual Illustration**: The concept of the down view is not entirely clear, especially concerning why it lacks color. A visual illustration showing the virtual camera position could enhance understanding.
2. **S-PR Explanation**: It’s challenging to grasp exactly how S-PR contributes to generation. A comparative figure demonstrating results before and after applying S-PR would clarify this aspect.
3. **Lack Real World Baseline**: The baseline experiment in real world is missing.
4. **Unclear Contribution of Different Designs**: Most of the performance in Table 2 (S-RVT2) are within 1 point, which makes it unclear whether many of them are still useful.

**Questions:**

1. **Down View Clarification**: Could the authors clarify the colorless nature of the down view and its virtual camera position? Including a figure illustrating the virtual camera position could make this aspect clearer. Also can you explain why the down view looks colorless compare with top view?
2. **Effect of S-PR on Generation**: The design of S-PR is unclear. Could the authors include a figure comparing the rendered images before and after applying S-PR? This would help illustrate S-PR’s impact on generation.
3. **Focused Experimentation in Table 2**:  The improvements mainly impact tasks needing precise top-down alignment (e.g., Insert Peg, Sort Shape). Given that most differences in Table 2 are within 1%, an experiment disabling both S-PR and down view could clarify other designs’ contributions. Additionally, statistical tests would help determine if small differences are meaningful. It will also be great if you can discuss the speed-accuracy trade-off applying these designs.
4. **Baseline and Failure Analysis**: Real-world experiments would benefit from a baseline comparison. A typical baseline would be an RVT/RVT2 without designs introduced in the paper. It would also help if the authors could conduct a failure analysis to highlight potential improvement areas. If challenges arose in implementing real-world baselines, an explanation would be valuable.
5. **Left View Performance Drop**: The results in Table 2 indicate a counterintuitive performance drop when incorporating a left view. A straightforward experiment replacing the right view with the left view (without adding extra views) could help isolate the cause. Based on this experiment, the authors can explore whether this drop is due to:
   - the presence of both left and right views introducing redundancy or conflicting information,
   - the left view alone, as opposed to the right view, negatively impacting performance,
   - or simply having an excess of views, which may complicate heatmap prediction?

---

> ### Author Response · Authors · 2024-11-21
> **To Reviewer jBkg 1**
>
> a) **W1 Lack of Visual Illustration**: The concept of the down view is not entirely clear, especially concerning why it lacks color. A visual illustration showing the virtual camera position could enhance understanding.
>
> **Q1 Down View Clarification**: Could the authors clarify the colorless nature of the down view and its virtual camera position? Including a figure illustrating the virtual camera position could make this aspect clearer. Also can you explain why the down view looks colorless compare with top view?
>
> **Q2 Effect of S-PR on Generation**: The design of S-PR is unclear. Could the authors include a figure comparing the rendered images before and after applying S-PR? This would help illustrate S-PR’s impact on generation.
>
> Thank you for pointing out the **lack of visual illustration**. The presentation in Figure 3 of the main text is not clear, and we have updated the manuscript. In the revised manuscript, we include a new figure that 1) illustrates the poses of virtual cameras (weakness 1), and 2) visualizes the rendered results from each virtual cameras using **PR** and **S-PR** respectively (weakness 2). This improved picture can be found in [To Reviewer jBkg](https://anonymous.4open.science/r/Super-Robot-View-Transformer-Rebuttal/) folder and also the revised manuscript. Regarding your question about why the *down view* lacks color, we would like to clarify that the rendered results do indeed contain color. We speculate that the limited field of view may have resulted in a relatively monotonous color range within that particular view, giving the impression of lacking color. We hope that the revised figure provides a clearer visual explanation.
>
> b) **W2 S-PR Explanation**: It’s challenging to grasp exactly how S-PR contributes to generation. A comparative figure demonstrating results before and after applying S-PR would clarify this aspect.
>
> We appreciate this suggestion on the issue of **S-PR explanation**. As we have discussed in a), we combine the two modifications you suggest into one picture in [To Reviewer jBkg](https://anonymous.4open.science/r/Super-Robot-View-Transformer-Rebuttal/) folder.

---

> ### Author Response · Authors · 2024-11-21
> **To Reviewer jBkg 2**
>
> c) **W3 Lack Real World Baseline**: The baseline experiment in real world is missing.
>
> **Q4 Baseline and Failure Analysis**: Real-world experiments would benefit from a baseline comparison. A typical baseline would be an RVT/RVT2 without designs introduced in the paper. It would also help if the authors could conduct a failure analysis to highlight potential improvement areas. If challenges arose in implementing real-world baselines, an explanation would be valuable.
>
> Thank you for highlighting the oversight of **missing real world baseline** and **failure analysis**. To address this, we perform additional real-world experiments using the original RVT/RVT2 models as baselines. The results are as follows:
>
> | Task                 | RVT  | RVT-2  | S-RVT(ours) | S-RVT2(ours) |
> | -------------------- | ---- | ------ | ----------- | ------------ |
> | Put item in drawer   | 30 % | 50 %   | 50 %        | **60 %**     |
> | Stack blocks         | 70 % | 70 %   | 70 %        | **80 %**     |
> | Place fruit on plate | 70 % | 80 %   | 80 %        | **90 %**     |
> | Plug charger         | 10 % | 70 %   | 60 %        | **80 %**     |
> | All tasks            | 45 % | 67.5 % | 65 %        | **77.5 %**   |
>
> From the results above, our S-RVT achieves notable enhancement over RVT in high-precision manipulation tasks, such as the *plug charger* task, with success rate improvements of 50%. We have updated the results in the supplementary materials.
>
> For the **failure analysis**, in fact, we have discussed failure cases in the last part of the paper, analyzing **failures in real-world scenarios** from two aspects. First, because only a single third-person view camera is deployed in the real environment, task-related objects may become completely occluded during execution. Second, due to the limited number of human demonstrations, the learned features may not be sufficiently discriminative.
>
> For the **failure cases in simulation tasks**, we identify two main sources of failure:
>
> 1. **Long-horizon decision-making.** Because the RVT model does not incorporate past observations, it may struggle to accurately determine its current state and that of the environment, leading to failures in executing tasks that require memory of previous steps.
> 2. **Tasks requiring large-angle rotations and precise alignments** (*e.g.*, hanging a mug on a mug tree). For example, in the *Place Cups* task shown in Table 1, the difficulty arises from:
>    1. Predicting the correct rotation (aligning the mug's handle with the "branches" of the mug tree at an appropriate angle).
>    2. Ensuring that the robotic arm's trajectory, obtained through inverse kinematics (IK) to reach the predicted key pose of the end effector, does not collide with the mug tree.
>
> These challenges are limitations of our method, but they also highlight possible directions for future work. For better visualization, we provide some failure cases in [To Reviewer JBkg](https://anonymous.4open.science/r/Super-Robot-View-Transformer-Rebuttal/README.md)
>
> We acknowledge that our discussion on future work is insufficient. We have updated the corresponding section in the revised supplementary materials as follows:
>
> *In future work, we aim to address camera occlusion in real-world scenarios from both temporal and spatial perspectives. Temporally, since objects are not continuously occluded throughout the sequence, enabling S-RVT to retain memory of past observations is valuable. Spatially, deploying multiple cameras from different viewpoints help mitigate occlusion. Regarding the limited number of demonstrations, generating additional demonstrations from synthetic data could enable the training of a robust robot using only a few real-world demos.*

---

> ### Author Response · Authors · 2024-11-21
> **To Reviewer jBkg 3**
>
> d) **W4 Unclear Contribution of Different Designs**: Most of the performance in Table 2 (S-RVT2) are within 1 point, which makes it unclear whether many of them are still useful.
>
> **Q3 Focused Experimentation in Table 2**: The improvements mainly impact tasks needing precise top-down alignment (e.g., Insert Peg, Sort Shape). Given that most differences in Table 2 are within 1%, an experiment disabling both S-PR and down view could clarify other designs’ contributions. Additionally, statistical tests would help determine if small differences are meaningful. It will also be great if you can discuss the speed-accuracy trade-off applying these designs.
>
> **Q4 Left View Performance Drop**: The results in Table 2 indicate a counterintuitive performance drop when incorporating a left view. A straightforward experiment replacing the right view with the left view (without adding extra views) could help isolate the cause. Based on this experiment, the authors can explore whether this drop is due to:
>
> - the presence of both left and right views introducing redundancy or conflicting information,
> - the left view alone, as opposed to the right view, negatively impacting performance,
> - or simply having an excess of views, which may complicate heatmap prediction?
>
> Thank you for your insightful comments regarding **Table 2**. Following your suggestions, we have added two sets of experiments: **1) removal of both the down view and S-PR** and **2) replacement of the original right view with the left view**. Due to space limitations, we present the results of these additional experiments in the table below.
>
> | S-PR          | Front         | Top           | Right         | Left          | Down          | Avg. Succ. |
> | ------------- | ------------- | ------------- | ------------- | ------------- | ------------- | ---------- |
> | $\checkmark $ | $\checkmark $ | $\checkmark $ | $\checkmark $ | $\times$      | $\checkmark $ | 0.878      |
> | $\times$      | $\checkmark $ | $\checkmark $ | $\checkmark $ | $\times$      | $\times$      | 0.824      |
> | $\checkmark $ | $\checkmark $ | $\checkmark $ | $\times$      | $\checkmark $ | $\checkmark $ | 0.877      |
>
> For **Exp 1**, as you point out, the S-PR and the down view are indeed the critical components that significantly impact the performance of RVT2. Enhancements such as super-resolution or HSP have minimal effect on RVT2. This is because RVT2 already employs a *coarse-to-fine* two-layer MVT structure, which effectively realizes the benefits of super-resolution and HSP in the output of the second layer. However, the two-layer MVT introduces additional parameters and computational complexity, and this cascading structure means that errors occurring in the first layer will propagate to the second layer. In comparison, our designed S-MVT and HSP structures more directly increase the resolution of the action space. Moreover, RVT2's performance on the RLBench benchmark is approaching saturation, making further improvements quite challenging.
>
> In response to your suggestion to discuss the speed-accuracy trade-off, we have included additional experiments detailing the training time and inference speed, as shown in the table below:
>
> | Model  | Training Time (days) | Inference Speed (fps) | Avg. Success |
> | ------ | -------------------- | --------------------- | ------------ |
> | RVT    | 0.85                 | 20.9                  | 0.629        |
> | RVT2   | 0.92                 | 20.1                  | 0.814        |
> | S-RVT  | 0.88                 | 20.5                  | 0.734        |
> | S-RVT2 | 1.03                 | 19.2                  | 0.878        |
>
> As observed, there is a slightly decrease in training time and inference speed compared to RVT and RVT-2. However, we believe that the improvements in accuracy provide greater value in terms of the speed-accuracy trade-off.
>
> For **Exp 2**, as shown in former table, replacing the right view with the left view results in essentially unchanged model performance. This indicates that the two views are equivalent. Therefore, the performance drop observed when adding the left view is likely due to the presence of both left and right views introducing redundancy or conflicting information.

---

### Author Response · Authors · 2024-11-25
**Request for Further Discussion on Review Comments**

Dear Reviewers,

Thank you for your previous feedback and time spent reviewing our submission. We truly appreciate the effort and insights provided. As we continue refining the work based on your comments, we believe further discussion or clarification on certain points would be beneficial.

Your input would greatly help in enhancing the quality and clarity of the work. We look forward to hearing from you and would be grateful for your further guidance.

Thank you once again for your time and assistance.

---

### Meta-Review · Area_Chair_2QVM · 2024-12-24

**Metareview:**

This paper proposes S-RVT, an extension of RVT-1/2 for 3D object manipulation. It aims to address limitations in RVT, such as occlusion artifacts and limited action resolution, by introducing components like the Super Point Renderer and the Hierarchical Sampling Policy. Experiments are conducted in both simulation and real-world settings (as discussed in the rebuttal).

Strengths:

- Significant empirical improvements over RVT on the RLBench benchmark.
- Multiple experiments and ablations in sim and real.

Weaknesses:

- As Reviewer jBkg pointed out, it is unclear whether all the new components are necessary, given the small percentage improvement in performance. Consequently, it is uncertain if all components contribute to a statistically significant improvement, especially since the authors mention "normal performance fluctuation" on the benchmark in the rebuttal.
- The explanation regarding S-RVT reducing uncertainty appears confusing and lacks theoretical justification. According to the AC, the paper would benefit from emphasizing its empirical strengths rather than making auxiliary claims about uncertainty without substantial evidence.

Overall:
While the paper demonstrates clear empirical gains in addressing the problem of 3D object manipulation, the necessity of all the introduced components remains unclear. Additionally, the focus on claims about uncertainty lacks clarity and theoretical grounding. Therefore, I recommend rejection of the current version.

**Additional Comments On Reviewer Discussion:**

During the rebuttal, some concerns regarding the absence of real-world experiments and the lack of clarity in explaining the various components were addressed. Following the rebuttal, two reviewers favored acceptance, while three leaned toward rejection.
The AC acknowledges that the paper has the potential to make a significant empirical contribution to 3D manipulation. However, the current presentation falls short in addressing the issues outlined above. These weaknesses played a major role in the decision to reject the current version of the paper.

---

### Decision · Program_Chairs · 2025-01-22

Reject